# Exploring Learning Complexity for Efficient Downstream Dataset Pruning

**Wenyu Jiang**[1,2]*, **Zhenlong Liu**[1], **Zejian Xie**[1], **Songxin Zhang**[1], **Bingyi Jing**[1], **Hongxin Wei**[1]†

[1]Department of Statistics and Data Science, Southern University of Science and Technology
[2]State Key Laboratory for Novel Software Technology, Nanjing University

## Abstract

The ever-increasing fine-tuning cost of large-scale pre-trained models gives rise to the importance of dataset pruning, which aims to reduce dataset size while maintaining task performance. However, existing dataset pruning methods require training on the entire dataset, which is impractical for large-scale pre-trained models. In this paper, we propose a straightforward, novel, and training-free hardness score named Distorting-based Learning Complexity (**DLC**), to identify informative images and instructions from the downstream dataset efficiently. Our method is motivated by the observation that easy samples learned faster can also be learned with fewer parameters. Specifically, we define the Learning Complexity to quantify sample hardness and utilize a lightweight weights masking process for fast estimation, instead of the costly SGD optimization. Based on DLC, we further design a flexible under-sampling strategy with randomness (dubbed **FlexRand**), replacing the top-K strategy, to alleviate the severe subset distribution shift. Extensive experiments with downstream image and instructions dataset pruning benchmarks demonstrate the effectiveness and efficiency of the proposed approach. In the images pruning benchmark, DLC significantly reduces the pruning time by **35×** while establishing *state-of-the-art* performance with FlexRand.

## 1 Introduction

The paradigm of pre-training and fine-tuning (PT-FT) (Kornblith et al., 2019) is increasingly popular with the rapid advancements in foundation models. Instead of training from scratch, the PT-FT paradigm first pre-trains a large and general model on broad data, and then fine-tuning the pre-trained model for a variety of downstream tasks. This is particularly advantageous for employing deep models on edge devices (e.g., telephone) (Dhar et al., 2021), where data privacy is crucial and training from scratch is infeasible due to limited computation resources. Unfortunately, on-device fine-tuning costs can escalate sharply with the increasing scale of data, due to the enormous size of pre-trained models[1]. This highlights the importance of downstream dataset pruning, which aims to reduce the dataset size for fine-tuning while maintaining the performance on downstream tasks.

In the literature, existing dataset pruning methods are generally designed for training from scratch. Those methods quantify the sample importance through a scoring function, which usually requires training the model on all the candidates. For example, EL2N (Paul et al., 2021) calculates the expected norm of loss gradients by training for multiple trials, sometimes even longer than the training time on large-scale datasets (Qin et al., 2023). When transferring to the PT-FT paradigm, the training or fine-tuning of the entire dataset can be prohibitively expensive and even infeasible as pre-trained models generally have huge parameters. Additionally, current methods typically do not take advantage of the pre-trained model, which is accessible prior to the fine-tuning and potentially impacts the selection of optimal data subset. These concerns motivate us to explore efficient dataset pruning methods that leverage pre-trained models without relying on backpropagation.

In this work, we propose a training-free hardness score, implementing the Learning Complexity (Definition 2) by a lightweight pre-training parameters masking process. From the hardness per-

---

*Work done at SUSTech as a visiting scholar.

†Corresponding author (`weihx@sustech.edu.cn`)

[1]https://epochai.org/data/notable-ai-models

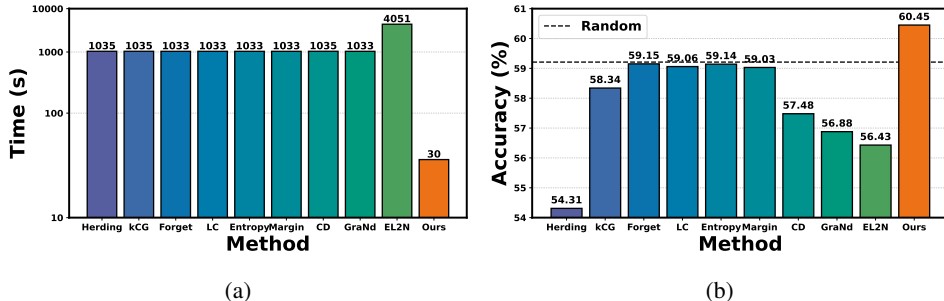

(a)        (b)

Figure 1: Performance comparison of different dataset pruning methods. **(a)**: Time for downstream dataset pruning. The costs of existing training-based methods are expensive, but we achieve **35×** speed up. **(b)**: Accuracy on the downstream task. Our method outperforms the random baseline and achieves *state-of-the-art* performance. More results can be found in Section 4.

spective, the above learning complexity aims to quantify sample importance via calculating the classification loss integral over a model sequence with ever-increasing classification performance, which is defined as the Learning Path (Definition 1). The proposed scoring function, Distorting-based Learning Complexity (dubbed **DLC**), is an efficient learning complexity implementation, by the observation that distorting (masking) the pre-training weights can construct a learning path without training or fine-tuning. Specifically, we mask pre-training weights multiple times with random ratios, and the learning complexity can be efficiently approximated by the Monte Carlo method (Caflisch, 1998). As a result, DLC can distinguish easy samples from hard ones. To further verify the effectiveness of DLC, we empirically demonstrate **a high ranking correlation** between DLC and the common optimization-based learning complexity. Moreover, we extend DLC to quantify the instruction hardness for efficiently fine-tuning the large language models (LLMs).

To identify informative samples based on DLC, we design a flexible under-sampling strategy with randomness, named **FlexRand**, replacing the common top-K strategy. For the multinomial distribution of under-sampling, FlexRand aims to flexibly adjust sampling preference at different data regimes. Moreover, the subset from FlexRand should not be significantly shifted away from the original data distribution. Specifically, we randomly select the same number of samples from the easy and hard intervals respectively, whose lengths are flexibly tuned with a splitting hyperparameter $\gamma$ to accommodate different data regimes. In this way, we have the following two advantages: **(a)**: FlexRand can adapt to different data regimes; **(b)**: FlexRand avoids severe distribution shift.

To verify the efficacy and efficiency of our method, we conduct extensive experiments on diverse setups, including different pre-training paradigms, model architectures, fine-tuning datasets, and pruning ratios. Empirical results show that our method establishes *state-of-the-art* performance over existing methods for dataset pruning. For example, our approach improves the downstream accuracy averaged over various setups from 59.21% to 60.45% - a **1.24%** improvement over the random method. Meanwhile, DLC significantly reduces the pruning time by **35×** as shown in Figure 1a. On the contrary, the compared methods achieve comparable or even worse performance than the random as shown in Figure 1b. For LLMs fine-tuning, our method consistently outperforms the random with various pre-trained models and instruction fine-tuning datasets. In addition, our analysis indicates that DLC is not sensitive to the pre-trained model, and the proposed sampling principle works well with different implementations of learning complexity.

## 2 PRELIMINARIES

### 2.1 BACKGROUND

**Setup.** In this paper, we consider the setting of supervised multi-class classification with a pre-trained encoder. Let $\mathcal{X} \subset \mathbb{R}^d$ denote the input space and $\mathcal{Y} = \{1, ..., K\}$ denote the corresponding label space. The downstream dataset $\mathcal{D} = \{(\boldsymbol{x}_i, y_i)\}_{i=1}^N$ is drawn *i.i.d* from the joint data distribution $\mathbb{P}_{\mathcal{X} \times \mathcal{Y}}$. We use $h$ to denote the pre-trained encoder and $g$ to denote the prediction head. Given the

downstream dataset, we train a classifier $f = g \circ h : \mathcal{X} \mapsto \mathbb{R}^{|\mathcal{Y}|}$ with learnable parameters $\boldsymbol{\theta} \in \mathbb{R}^p$, which maps an input to the label space. An ideal classifier $f_\theta$ can be obtained by minimizing the following expected risk:

$$\mathcal{R}_\mathcal{L}(f) = \mathbb{E}_{(\boldsymbol{x},y) \sim \mathcal{P}_{\mathcal{X} \times \mathcal{Y}}}[\mathcal{L}(f(\boldsymbol{x}; \boldsymbol{\theta}), y)],$$

In practice, we optimize the classifier by minimizing the following empirical risk:

$$\mathcal{R}_\mathcal{L}^{\mathrm{emp}}(f, \mathcal{D}) = \frac{1}{N} \sum_{i=1}^{N} [\mathcal{L}(f(\boldsymbol{x_i}; \boldsymbol{\theta}), y_i)],$$

where $\mathcal{L}$ is the commonly used cross-entropy loss with the softmax activation function. Let $\tilde{z}$ and $\hat{z}$ denote the feature of $\boldsymbol{x}$ from the initial and fine-tuned $h$ respectively.

**Problem statement.** The over-parameterized networks can be too heavy to optimize with limited computing resources. To accommodate the training budget $\eta \in (0, 1)$ for downstream tasks, the dataset pruning aims to select a subset $\hat{\mathcal{D}} = \{(\boldsymbol{x}_i, y_i)\}_{i=1}^{M} \subset \mathcal{D}$ $(M \leq \eta * N)$, which can be used to train a classifier $\hat{f}$ by minimizing the empirical risk $\mathcal{R}_\mathcal{L}^{\mathrm{emp}}(f, \hat{\mathcal{D}})$. The classifier from the ideal subset should have the minimal expected risk $\mathcal{R}_\mathcal{L}(\hat{f})$. This can be formulated as a bilevel optimization problem with a cardinality constraint:

$$\min_{\hat{\mathcal{D}}} \quad \mathcal{R}_\mathcal{L}(\hat{f})$$
$$\text{s.t.} \quad \hat{f} = \arg\min_\theta \mathcal{R}_\mathcal{L}^{\mathrm{emp}}(f, \hat{\mathcal{D}})$$
$$|\hat{\mathcal{D}}| \leq \eta * N$$

Enumerating and evaluating all the $\binom{N}{M}$ subsets is non-trivial. An alternative solution performs the dataset pruning by a level-set estimation:

$$r((\boldsymbol{x}, y)) = \begin{cases} \text{pruning}, & \text{if } \mathrm{S}((\boldsymbol{x}, y)) < \tau_y \\ \text{preserving}, & \text{if } \mathrm{S}((\boldsymbol{x}, y)) \geq \tau_y \end{cases}$$

where $\mathrm{S}((\boldsymbol{x}, y))$ denotes a scoring function to quantify sample importance and $\tau_y$ is a threshold specified to accommodate the budget $\eta * N/K$ for category $y$. By convention, the common under-sampling strategy keeps the **top-K** samples sorted by the $\mathrm{S}((\boldsymbol{x}, y))$ in descending order. However, existing scoring functions typically rely on fine-tuning the pre-trained model on the entire downstream dataset (Coleman et al., 2020), which is undesirable for efficient edge fine-tuning. To reduce downstream dataset pruning costs, we propose an efficient scoring function solely based on the pre-trained model without fine-tuning in the following section.

## 3 METHOD

To design an efficient scoring function, we first define the Learning Complexity to quantify sample importance from the hardness perspective. Specifically, we calculate the integral of classification loss for a sample predicted by an improving model sequence, which is defined as:

**Definition 1 (Learning Path)** *A sequence of model parameters* $\boldsymbol{\Theta} = \{\boldsymbol{\theta}(t) \mid t \in \mathcal{T}\}$ *can be defined as a learning path if there exists a positive constant* $r < \mathcal{R}_\mathcal{L}(f_{\boldsymbol{\theta}(0)})$ *such that* $\lim_{t \to \infty} \mathcal{R}_\mathcal{L}(f_{\boldsymbol{\theta}(t)}) = r$.

Thus, the above Learning Complexity can be formally defined as follows:

**Definition 2 (Learning Complexity)** *For any training sample* $(\boldsymbol{x}, y) \in \mathcal{D}$*, given a learning path* $\boldsymbol{\Theta}$*, the learning complexity is defined as*

$$\mathrm{S}_{\mathrm{LC}}((\boldsymbol{x}, y)) = \int_{t \in \mathcal{T}} \mathcal{L}(f(\boldsymbol{x}; \boldsymbol{\theta}(t)), y) dt$$

Intuitively, samples with low learning complexity mean correct classification by a weak classifier in the front part of the learning path, compared to the hard ones.

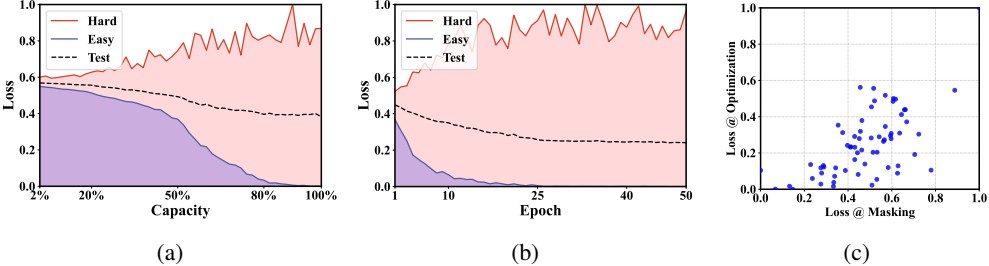

(a)  (b)  (c)

Figure 2: Ranking correlation between the loss integral over the optimization and masking process. We fine-tune the pre-trained ResNet-18 on the five downstream datasets for 50 epochs and present the **CXRB10** (Nodule) results here due to the space limit (see Figure 7 for full results). The easy (hard) sample corresponds to the data point with a minimum (maximum) loss integral over the optimization. We utilize Max-Min normalization to scale the loss to the range of $(0, 1)$. **(a)**: Loss trends with the number of parameters, i.e., model capacity. We produce models with different capacities via masking $\{2\%, ..., 98\%\}$ weights of the pre-trained ResNet-18. Average category representation acts as the prototype for classification and loss integral. **(b)**: Loss trends with the optimization time. **(c)**: High ranking correlation coefficient with $\rho = \mathbf{0.54}$.

## 3.1 DLC: DISTORTED-BASED LEARNING COMPLEXITY

In this part, we propose an efficient scoring function, Distorting-based Learning Complexity (dubbed **DLC**) by implementing the above learning complexity with a lightweight distorting process. Concretely, we distort the pre-trained model by masking the initial weights with different ratios, and the resulting models constitute an effective learning path. For clarity of expression, we refer to distorting as masking in the remaining part.

**Efficiently constructing the learning path with masking**  DLC is motivated by the observation that increasing the number of parameters leads to improved generalization (Valle-Perez et al., 2019; Neyshabur et al., 2018; 2017). Although the pre-trained models have not been fine-tuned on the downstream datasets, the expected risk (test error) on the target task still decreases with the increasing model capacity as shown in Figure 2a. Formally, when the capacity of pre-trained model $f_{\hat{\theta}}$ at step $t + 1$ is larger than that at step $t$, we have $\mathcal{R}_{\mathcal{L}}(f_{\hat{\theta}(t+1)}) < \mathcal{R}_{\mathcal{L}}(f_{\hat{\theta}(t)})$. Therefore, the sequence of pre-trained models with different capacities is viable and can be constructed efficiently without fine-tuning. Specifically, we directly produce the classifiers with different capacities by masking the initial pre-training weights with ratios ranging from $\{2\%, 4\%, ..., 98\%, 100\%\}$. Note that parameters with small L1 norm are masked first (Han et al., 2015). We provide a concrete formulation of the pre-training weights masking in Appendix B. To calculate the loss integral, i.e., Learning Complexity, the average category feature acts as the class prototype, and we further adopt the Monte Carlo method to approximate the loss integral (Caflisch, 1998) for efficiency, instead of enumerating all models with different capacities:

$$\mathrm{S_{LC}}((\boldsymbol{x}, y)) \approx \frac{|\mathcal{T}|}{|T|} \sum_{t \in T} \mathcal{L}(f(\boldsymbol{x}; \boldsymbol{\theta}(t)), y)$$

where $T$ comprises random points $t$ from $\mathcal{T}$. In this way, DLC can distinguish easy samples from the hard ones as shown in Figure 3. To further quantitatively verify the effectiveness of DLC, we implement the common optimization-based learning complexity and demonstrate **a high ranking correlation between those two implementations**.

**Optimization trajectories as a viable but costly learning path**  We first show that the optimization trajectories constitute a viable learning path. At the training stage, stochastic gradient descent (SGD) and its variants (Sutskever et al., 2013; eon Bottou, 1998) are usually used for optimizing the $\mathcal{R}_{\mathcal{L}}^{\mathrm{emp}}(f, \mathcal{D})$ starting from the initial $\boldsymbol{\theta_0}$. The optimization trajectories can be formulated with the following updating rule:

$$\boldsymbol{\theta}_{t+1} = \boldsymbol{\theta}_t - \alpha_t \left( \frac{1}{|\mathcal{B}_t|} \sum_{(\boldsymbol{x_i}, y_i) \in \mathcal{B}_t} \nabla[\mathcal{L}(f(\boldsymbol{x_i}; \boldsymbol{\theta}_t), y_i)] \right)$$

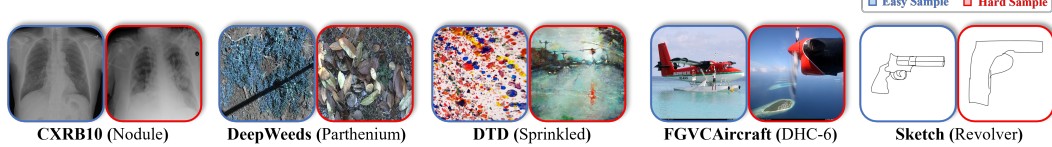

Figure 3: Samples distinguished by DLC. In detail, we randomly select five image pairs with the highest and lowest DLC scores from all downstream **Dataset** (Category) sets. Qualitatively, easy samples with the lowest DLC contain a full and clear structure for classification.

where $\mathcal{B}_t$ is the data batch sampled from $\mathcal{D}$ and $\alpha_t$ is the learning rate at iteration $t$ ($t$ is discrete and starts from 0). Despite not guaranteeing convergence to a global optimum, SGD tends to find the local minimum $\hat{\boldsymbol{\theta}}$ with flatness (Li et al., 2018; Keskar et al., 2017; Hardt et al., 2016) such that

$$\lim_{t \to \infty} \mathcal{R}_{\mathcal{L}}(f_{\boldsymbol{\theta}(t)}) = \mathcal{R}_{\mathcal{L}}(f_{\hat{\boldsymbol{\theta}}}) = r < \mathcal{R}_{\mathcal{L}}(f_{\boldsymbol{\theta}(\mathbf{0})})$$

To empirically verify the asymptotic convergence and decreasing generalization error, we visualize the test loss trend in Figure 2b. Obviously, $f_{\boldsymbol{\theta}(0)}$ gradually converges to $f_{\hat{\boldsymbol{\theta}}}$ with better generalization performance. Following the Definition 1, the optimization trajectory of SGD during the fine-tuning is a viable learning path, and we calculate the integral of classification loss over the fine-tuning process as the optimization-based learning complexity.

We denote the above different implementations of learning complexity as $S_{\text{Opt}}$ and $S_{\text{Mask}}$ respectively. Despite implementing $S_{\text{LC}}$ in different forms, we empirically demonstrate the strong ranking correlation between the two as shown in Figure 2c, which indicates the effectiveness of DLC. Specifically, the average Spearman's coefficient $\rho$ (Zar, 2005) between the $S_{\text{Opt}}$ and $S_{\text{Mask}}$ over different downstream datasets is as high as **0.51**. As a result, DLC can efficiently quantify the sample hardness without dependency on training or fine-tuning.

**An intuitive interpretation**   As shown in Figure 2b, we can observe that the $f$ can classify the easy example with a smaller loss at any iteration $t$. The same conclusion holds when the pre-trained model has fewer parameters as shown in Figure 2a. In other words, easy samples learned faster can also be learned with fewer parameters.

### 3.2   FLEXRAND: FLEXIBLE UNDER-SAMPLING WITH RANDOMNESS

From our previous analysis, we show that DLC can discriminate the hard samples from the easy. To identify informative samples with DLC, we further design a flexible under-sampling strategy with randomness, named **FlexRand**, replacing the common top-K strategy. For the multinomial distribution of under-sampling, FlexRand aims to flexibly adjust sampling preference at different data regimes. On the other hand, the subset from FlexRand should not be significantly shifted away from the original data distribution. To achieve these objectives, we randomly select the same number of samples from the easy and hard intervals respectively, whose lengths are flexibly tuned with a splitting hyperparameter $\gamma$ to accommodate different data regimes. The sampling probability for $(\boldsymbol{x}, y)$ can be formulated as follows:

$$p((\boldsymbol{x}, y)) = \begin{cases} \frac{M}{2N * \gamma}, & \text{if } S_{\text{LC}}((\boldsymbol{x}, y)) < S_\gamma \\ \frac{M}{2N * (1-\gamma)}, & \text{if } S_{\text{LC}}((\boldsymbol{x}, y)) \geq S_\gamma \end{cases}$$

where $S_\gamma$ is the $\gamma$-percentage of sorted samples scores, $N$ is the dataset size and $M$ is the subset size. In this way, we have the following two advantages:

**FlexRand can adapt to different data regimes.**   In a data-poor regime, easy samples are more informative, and the opposite holds when more samples can be preserved for fine-tuning (Sorscher et al., 2022). However, the top-K (Easy or Hard) strategy, denoted as T-E or T-H, fixes sampling preference across different data regimes and shows inferior downstream performance compared to the random as shown in Figure 4b & 4d. On the contrary, FlexRand can flexibly adjust sampling preference with a splitting hyperparameter $\gamma$ to accommodate different data regimes.

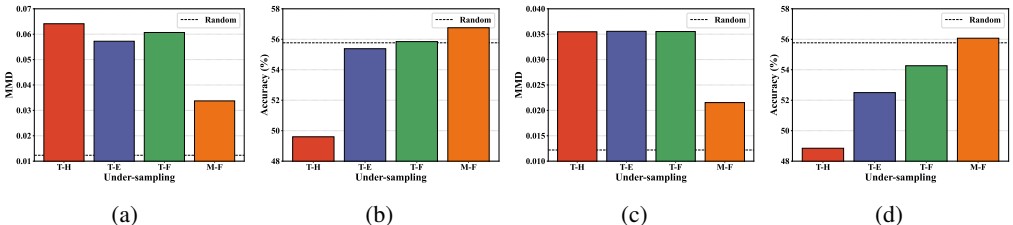

Figure 4: Harmful distribution shift from the top-K under-sampling strategies. We conduct downstream dataset pruning with four under-sampling strategies according to the optimization-based learning complexity and DLC. **(a)**: Distribution shift comparison of four subsets for optimization-based learning complexity. **(b)**: Downstream performance comparison of four subsets for optimization-based learning complexity. **(c)**: Distribution shift comparison of four subsets for DLC. **(d)**: Downstream performance comparison of four subsets for DLC.

**FlexRand avoids severe distribution shift.** Due to the importance of flexible sampling preference, we change the fixed top-K strategy to accommodate different data regimes. Despite improvement over the original, its performance still lags behind the random under-sampling as shown in Figure 4b & 4d. To analyze and investigate such performance degradation from the top-K (Flexible) strategy (denoted as T-F), we compare the distribution distance of different subsets to the full datasets with maximum mean discrepancy (MMD) (Gretton et al., 2012). As shown in Figure 4a & 4c, the top-K sampling incurs severe distribution shifts compared with the random, which deteriorates downstream performance. On the contrary, FlexRand avoids such distribution shift by randomly selecting the same number of samples from the easy and hard intervals respectively. When mixing the top-K (Flexible) with the random strategy (denoted as M-F), corresponding to the extreme splitting $\gamma$ values, we can find that the distribution shift is still mild with better downstream performance as shown in Figure 4b & 4d.

### 3.3 EXTENSION TO INSTRUCTION DATASET PRUNING

Recently, large language models (LLMs) (Touvron et al., 2023a;b; Jiang et al., 2024) driven by the PT-FT paradigm have shown incredible capabilities across a wide range of language tasks. Despite the availability of pre-trained weights, the ever-increasing number of parameters still incur heavy computation overhead for the fine-tuning, which involves instruction-tuning (Ouyang et al., 2022) and reinforcement learning with human feedback (RLHF) (Knox & Stone, 2011) to align the pre-trained LLMs with human preferences. On the other hand, recent works (Zhou et al., 2023a; Cao et al., 2023) indicate that almost all knowledge in LLMs is learned during pre-training, and only limited instruction tuning data is necessary to teach models to produce high-quality output. Therefore, we further extend the learning complexity to prune instruction datasets for efficient large language models fine-tuning (Dubey et al., 2024; Team et al., 2024). Specifically, we construct the learning path by masking different numbers of weights in the pre-trained large language models (LLMs), which are sorted by L1 norm in ascending order. Due to the prediction difference in image classification and text generation, we replace the original loss function $\mathcal{L}(f(\boldsymbol{x}; \boldsymbol{\theta}, y)$ with:

$$\mathrm{S}_{\mathrm{LC}}((\boldsymbol{x}, \boldsymbol{y})) = \frac{1}{C} \sum_{j=0}^{C-1} \mathcal{L}(f(y_{j-1} : ... : y_0 : \boldsymbol{x}; \boldsymbol{\theta}(t)), y_j)$$

where $C$ is the length of the output $\boldsymbol{y}$.

## 4 EXPERIMENTS

In this section, we present the diverse downstream image and instruction datasets pruning benchmarks and empirically validate the effectiveness and efficiency of the proposed method. Moreover, we perform ablation studies to understand better how different components and hyperparameters influence the performance. The code is available in the supplementary material.

Table 1: Average classification accuracy over **5** diverse downstream datasets and pruning ratios with varying model architectures. **Bold** numbers are the optimal results, and underline numbers are suboptimal results. Detailed results can be found in Appendix C.2.2.

| Method | Accuracy (%) ↑ | | | | | Time (s) ↓ | |
|---|---|---|---|---|---|---|---|
| | **RN18** | **RN50** | **ViT-S** | **ViT-B** | **Average** | S($(\boldsymbol{x}, y)$) | **Total** |
| Random | 55.92 ± 0.21 | 60.22 ± 0.03 | 61.40 ± 0.01 | 59.31 ± 0.25 | 59.21 ± 0.12 | - | - |
| Herding | 48.25 ± 0.88 | 52.26 ± 0.11 | 59.53 ± 0.33 | 57.19 ± 0.57 | 54.31 ± 0.42 | 1035.0 ± 3.8 | 1035.0 ± 3.8 |
| kCG | 54.31 ± 0.35 | 58.33 ± 0.13 | 61.56 ± 0.11 | 59.16 ± 0.20 | 58.34 ± 0.14 | 1035.2 ± 4.4 | 1035.2 ± 4.4 |
| Forgetting | 55.46 ± 0.11 | 59.77 ± 0.10 | 61.94 ± 0.19 | 59.45 ± 0.25 | 59.15 ± 0.11 | 1033.0 ± 1.6 | 1033.0 ± 1.6 |
| Least Conf | 56.44 ± 0.27 | 60.38 ± 0.03 | 60.38 ± 0.15 | 59.05 ± 0.12 | 59.06 ± 0.01 | 1033.0 ± 0.9 | 1033.0 ± 0.9 |
| Entropy | 56.56 ± 0.30 | 60.53 ± 0.12 | 60.35 ± 0.46 | 59.14 ± 0.26 | 59.14 ± 0.07 | 1033.0 ± 1.4 | 1033.0 ± 1.4 |
| Margin | 56.40 ± 0.26 | 60.19 ± 0.08 | 60.42 ± 0.24 | 59.10 ± 0.36 | 59.03 ± 0.07 | 1033.0 ± 0.8 | 1033.0 ± 0.8 |
| CD | 53.54 ± 0.14 | 58.30 ± 0.11 | 60.70 ± 0.25 | 57.36 ± 0.08 | 57.48 ± 0.02 | 1035.3 ± 6.2 | 1035.3 ± 6.2 |
| GraNd | 52.67 ± 0.0 | 57.62 ± 0.06 | 60.41 ± 0.40 | 56.83 ± 0.02 | 56.88 ± 0.11 | 1033.0 ± 5.7 | 1033.0 ± 5.7 |
| EL2N | 52.15 ± 0.14 | 57.10 ± 0.15 | 60.12 ± 0.34 | 56.34 ± 0.49 | 56.43 ± 0.04 | 4051.0 ± 8.7 | 4051.0 ± 8.7 |
| SSP | 56.72 ± 0.41 | 57.31 ± 0.13 | 60.90 ± 0.33 | 59.60 ± 0.26 | 58.63 ± 0.25 | 1033.0 ± 0.7 | 1033.0 ± 0.7 |
| **Ours** | **57.31** ± 0.22 | **61.45** ± 0.02 | **62.46** ± 0.12 | **60.59** ± 0.20 | **60.45** ± 0.04 | **21.5** ± 0.1 | **29.5** ± 0.1 |

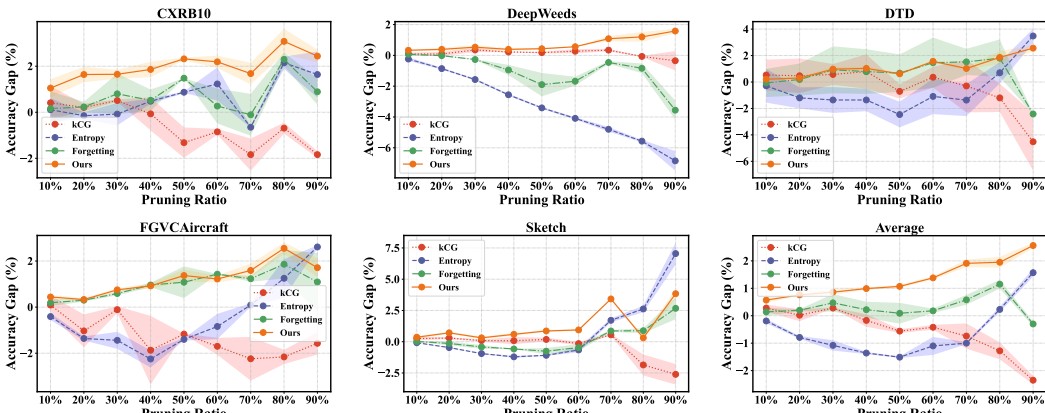

Figure 5: Trends of average accuracy gap over the random method at different pruning ratios. Detailed results can be found in Appendix C.2.2.

## 4.1 SETUP

### 4.1.1 DOWNSTREAM IMAGE DATASET PRUNING BENCHMARK

**Pre-trained encoders** For a comprehensive evaluation, we adopt pre-training encoders with different depths, architectures, and pre-training paradigms. In detail, the pre-trained encoders consist of ResNet-18, ResNet-50 (He et al., 2016), ViT-Small, and ViT-Base (Dosovitskiy, 2020). Those models are pre-trained on the canonical ImageNet-1K (Deng et al., 2009) dataset via fully and weakly supervised learning (Yalniz et al., 2019) respectively.

**Downstream datasets** Generally, the downstream dataset is domain-specific and different from the pre-training dataset ImageNet-1K. Therefore, we choose diverse downstream datasets from **5** domains (Islam et al., 2021) to construct the large-scale benchmark, including CXRB10[2], Deep-Weeds (Olsen et al., 2019), DTD (Cimpoi et al., 2014), FGVCAircraft (Maji et al., 2013), and Sketch (Eitz et al., 2012). For hyperparameter tuning, we split 20% as the validation set.

### 4.1.2 DOWNSTREAM INSTRUCTION DATASET PRUNING BENCHMARK

**Pre-trained models** We utilize 3 public pre-trained language models, including Mistral 7B (Jiang et al., 2023), Llama3 8B (Dubey et al., 2024), and Gemma2 9B (Team et al., 2024).

---

[2]CXRB10 is created by selecting 10 balanced classes from the ChestX-ray14 (Borghesi & Maroldi, 2020).

Table 2: Average classification accuracy (%) of Random / Ours methods on the MMLU benchmark. Different base models are fine-tuned with 50% instruction data from Alpaca Cleaned and Dolly & HH-RLHF, respectively. **Bold** numbers are the optimal results.

| Base Model | Alpaca Cleaned | | | | Dolly & HH-RLHF | | | |
|---|---|---|---|---|---|---|---|---|
| | **Humanity** | **Social Science** | **STEM** | **Other** | **Humanity** | **Social Science** | **STEM** | **Other** |
| **Mistral 7B** | 52.44 / **54.75** | 71.89 / **72.64** | 51.74 / **52.74** | 68.88 / **70.20** | 52.50 / **53.82** | 69.58 / **71.47** | 51.18 / **53.30** | 68.01 / **68.91** |
| **Llama3 8B** | 54.24 / **56.75** | 71.99 / **72.05** | 52.33 / **54.84** | 69.78 / **70.20** | 52.52 / **53.94** | 69.13 / **72.60** | 49.92 / **53.60** | 68.39 / **69.65** |
| **Gemma2 9B** | 56.37 / **58.79** | 73.29 / **75.25** | 54.14 / **56.30** | 71.13 / **71.52** | 55.21 / **56.08** | 71.43 / **73.32** | 50.23 / **52.99** | 69.51 / **70.60** |

**Instruction fine-tuning datasets** To align the based pre-trained models, we adopt 2 different instruction fine-tuning datasets: Alpaca Cleaned (Taori et al., 2023) and Dolly & HH-RLHF[3]. More setup details are presented in Appendix C.1.

## 4.2 MAIN RESULTS

### 4.2.1 FINE-TUNING FOR IMAGE CLASSIFICATION

**Our method achieves superior accuracy with much less time costs.** To investigate the effectiveness of dataset pruning methods with different pre-trained models, we evaluate the downstream accuracy for four architectures and present the results in Table 1. We find that the proposed method establishes *state-of-the-art* performance on all four architectures, highlighting its validity without dependency on model capacities and structure. On the contrary, the downstream accuracy of the most competitive baseline Forgetting deteriorates when changing the pre-trained encoder from ViT to ResNet. As a result, existing dataset pruning methods perform worse than the random strategy, while our method outperforms the random by **1.24%** on average.

In addition to the accuracy evaluated on different models and downstream setups, we also record the time cost of each baseline, including hyperparameter tuning associated with the scoring estimation and under-sampling. In Table 1, time results averaged over 5 downstream datasets are presented. Benefiting from the training-free scoring function and efficient hyperparameter tuning, the proposed method significantly reduces the pruning cost by **35×** compared with the one-shot fine-tuning baselines while achieving *state-of-the-art* downstream performance. Notably the acceleration is more remarkable with ever-increasing model capacities. Therefore, our method gives rise to the efficient fine-tuning of the over-parameterized pre-trained models.

**Our method works well at various pruning ratios.** For efficacy verification of our method on diverse downstream setups, we compare the accuracy gap trends over the random strategy for each fine-tuning dataset. As shown in Figure 5, the proposed method shows consistent improvement for each downstream dataset at different data regimes, indicating the efficacy of the flexible under-sampling strategy with randomness. Overall, the advantage is coherent and more evident when fine-tuning with fewer samples. For example, our method outperforms the random baseline by **2.6%** when removing 90% samples from the Sketch dataset. Despite the randomness in the custom under-sampling, our method demonstrates stability with smaller performance variance. However, the compared methods show inferior performance with fluctuating trends, which reveals the drawbacks of the fixed sampling principle and the challenge of diverse downstream dataset pruning benchmarks.

### 4.2.2 FINE-TUNING LLMs FOR TEXT CLASSIFICATION

In Table 2, we present the text classification results of different dataset pruning methods on the MMLU benchmark. We can find that the proposed method consistently outperforms the random method with various pre-trained models and instruction fine-tuning datasets. Such superiority further demonstrates that learning complexity is universal and applicable in visual and language domains.

## 4.3 ABLATION STUDIES

To better understand why our method can obtain superior efficiency and efficacy on the downstream dataset pruning task, we perform extensive ablation studies on the learning complexity metric, under-

---

[3]https://huggingface.co/datasets/mosaicml/dolly_hhrlhf

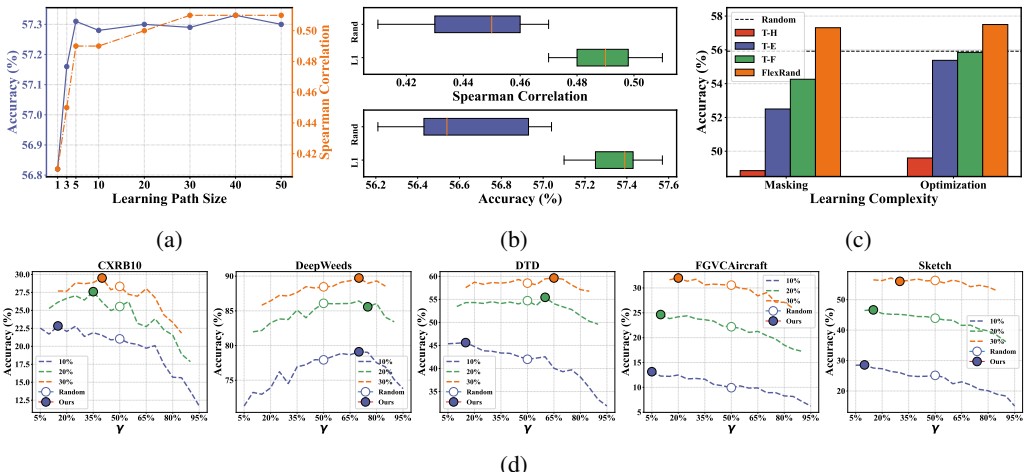

Figure 6: Results for different ablation studies. **(a)**: Ablation on the learning path size. **(b)**: Ablation on the masking strategies. **(c)**: Ablation on the under-sampling strategies. T-H, T-E, and T-F denote the top-K Hard, Easy, and Flexible under-sampling, respectively. **(d)**: Ablation on the splitting $\gamma$. Blue, green, and orange dotted lines correspond to 10%, 20%, and 30% of the data, respectively.

sampling strategy, and corresponding hyperparameters. We also present a comparison of FlexRand and Dataset Quantization (Zhou et al., 2023b) in Appendix C.2.4. As for the setups, we fine-tune the fully pre-trained ResNet-18 on 5 downstream datasets with 9 pruning ratios with default parameters.

**Learning path size**    In Figure 6a, we investigate the effect of the learning path size on the ranking correlation and downstream accuracy. In particular, we calculate the ranking correlation by the class-averaged Spearman correlation coefficients between the optimization-based learning complexity and masking-based. The results show the ranking correlation between the two scoring functions is significant and saturated at size 5. As a result, the proposed method can estimate the learning complexity efficiently and establish superior classification performance.

**Weight masking principle**    To elucidate the choice of weight masking principle, we compare the L1-based model pruning method with the random with 10 different seeds. Class-averaged Spearman correlation coefficients and accuracy are presented in Figure 6b. We find that randomly masking the pre-training weights incurs unstable ranking and inferior performance. On the contrary, the L1-based weight masking is deterministic with consistency and superiority across different seeds. Therefore, we choose to mask the weights according to the L1 norm.

**Under-sampling strategy**    As shown in Figure 6c, we show downstream accuracy with different under-sampling strategies to verify the effectiveness of FlexRand. FlexRand outperforms the random by **1.39%** with DLC. However, the easy and hard strategies lag far behind the random baseline. Compared with the flexible strategy, the improvement demonstrates the efficacy of the randomness, which alleviates the distribution shift. Besides, the optimization-based learning complexity with the FlexRand also established the *state-of-the-art* performance. This indicates that the proposed sampling principle works well with different implementations of learning complexity.

**Interval splitting**    To examine the effects of interval splitting, we set the $\gamma$ by traversing {0.05, 0.10, ..., 0.95}, and present the results in Figure 6d. We can find that small splitting values ($\gamma < 0.5$) generally deliver better performance in data-poor regimes. Therefore, our method outperforms the random baseline. While $\gamma$ tuned by the linear classifier does not always achieve the optimal, it significantly reduces the cost of hyperparameter tuning and establishes superior performance.

Table 3: Average accuracy (%) over **5** downstream datasets and varying pruning ratios with weakly pre-trained ResNet-18. **Bold** and underline numbers are the optimal and suboptimal results.

| Method | 10%~30% | | | | | 40%~60% | | | | | 70%~90% | | | | | Average |
|---|---|---|---|---|---|---|---|---|---|---|---|---|---|---|---|---|
| | CXR | DW | DTD | FA | Sk | CXR | DW | DTD | FA | Sk | CXR | DW | DTD | FA | Sk | |
| Random | 27.27 | 87.42 | 54.67 | 32.15 | 56.89 | 32.80 | 92.91 | 66.32 | 55.36 | 72.81 | 34.68 | 94.45 | 69.44 | 66.97 | 77.35 | 61.43 |
| Herding | 24.77 | 49.83 | 32.31 | 25.32 | 44.74 | 31.90 | 78.43 | 56.41 | 50.32 | 67.55 | 36.28 | 91.89 | 67.55 | 65.71 | 76.40 | 53.29 |
| kCG | 25.48 | 87.30 | 49.13 | 28.50 | 55.83 | 31.27 | 93.25 | 65.67 | 54.74 | 73.63 | 35.30 | 94.66 | 70.29 | 67.42 | 77.60 | 60.67 |
| Forgetting | 28.60 | 86.53 | 54.99 | **35.94** | **58.75** | 34.17 | 91.89 | 66.86 | 59.86 | 73.12 | 36.18 | 94.33 | 70.29 | 68.39 | 76.92 | 62.45 |
| Least Conf | 27.13 | 83.48 | 55.60 | 30.63 | 58.07 | 32.93 | 89.71 | 64.49 | 51.20 | 71.09 | 36.20 | 93.26 | 69.18 | 64.24 | 76.68 | 60.26 |
| Entropy | 28.42 | 82.95 | 55.52 | 31.09 | 58.45 | 32.88 | 89.93 | 64.41 | 51.28 | 71.23 | 35.72 | 93.55 | 68.86 | 64.27 | 76.80 | 60.36 |
| Margin | 28.12 | 83.20 | 55.80 | 30.52 | 58.43 | 33.92 | 89.85 | 64.21 | 50.59 | 71.35 | 36.52 | 93.41 | 68.63 | 64.30 | 76.61 | 60.36 |
| CD | 23.00 | 87.07 | 52.97 | 34.49 | 54.85 | 32.03 | 94.13 | 66.14 | 59.53 | 73.73 | 36.07 | 95.11 | 70.52 | 68.67 | 78.20 | 61.77 |
| GraNd | 21.77 | 86.40 | 53.43 | 35.08 | 54.24 | 30.70 | 94.04 | 66.02 | 60.38 | 73.38 | 36.03 | 94.85 | 70.35 | 68.91 | 78.00 | 61.57 |
| EL2N | 19.62 | 87.13 | 52.91 | 34.45 | 53.70 | 31.70 | 94.31 | 66.24 | 60.94 | 73.13 | 36.17 | 95.08 | 70.35 | 68.91 | 77.75 | 61.49 |
| Ours | **29.69** | **88.49** | **57.45** | 34.67 | 58.41 | **34.79** | 93.40 | **67.12** | 56.97 | 73.65 | **37.42** | 94.90 | **70.66** | 67.29 | 77.79 | **62.85** |

Table 4: Average classification accuracy (%) over **5** downstream datasets and **9** pruning ratios for different scores. **Bold** numbers are the optimal results, and underline numbers are suboptimal results.

| $S((\boldsymbol{x}, y))$ | ResNet-50 | | | | | ViT-Small | | | | | ViT-Base | | | | | Average |
|---|---|---|---|---|---|---|---|---|---|---|---|---|---|---|---|---|
| | CXR | DW | DTD | FA | Sk | CXR | DW | DTD | FA | Sk | CXR | DW | DTD | FA | Sk | |
| Random | 31.73 | 91.99 | 67.44 | 46.75 | 63.63 | 33.13 | 92.60 | 66.56 | 48.02 | 66.27 | 31.88 | 91.86 | 64.81 | 44.94 | 64.10 | 60.38 |
| Original | 32.90 | 92.50 | 68.29 | 47.61 | 64.48 | 34.12 | 92.99 | 67.29 | 48.41 | 66.93 | 32.63 | 92.17 | 65.26 | 45.63 | 65.06 | 61.08 |
| Transfer (RN18) | 31.61 | 90.43 | 68.03 | 46.78 | 63.98 | 34.71 | 92.93 | 67.45 | 48.37 | 66.69 | 32.94 | 91.82 | 65.12 | 44.99 | 64.61 | 60.70 |

# 5 DISCUSSION

**Is the proposed method affected by the quality of pretraining datasets?** While our method provides outstanding performances on models pre-trained by fully supervised learning, it is underexplored whether the proposed method can work well on those models pre-trained on low-quality data, i.e., weakly supervised learning. To investigate the effectiveness of our method in this scenario, we utilize the ResNet-18 model pre-trained on the dataset with missing labels (Yalniz et al., 2019). We present the classification accuracy of those pre-trained models by linear probing in Appendix C.2.5. As shown in Table 3, the proposed method keeps *state-of-the-art* performance and outperforms the random baseline by **1.42%** on average, not affected by the quality of the pre-training dataset. We provide detailed results in Appendix C.2.3.

**Can we use small models to select samples for large models?** For better efficiency in scoring estimation, we transfer the proposed scoring function from a pre-trained ResNet-18 to models with more parameters. The pruning time cost and accuracy averaged over 5 downstream datasets with 9 pruning ratios are presented in Table 4. Notably, its improvement over the random by **0.32%** implies the effectiveness of our method. However, we find that transferring does not achieve the original performance while consuming less time.

# 6 CONCLUSION

In this paper, we propose Distorted-based Learning Complexity (**DLC**), a novel and straightforward hardness score without relying on training or fine-tuning. We construct the learning path by distorting (masking) the initial pre-training weights with different ratios and calculate the loss integral via the Monte Carlo method for efficiency. Additionally, we design a flexible under-sampling strategy with randomness, named **FlexRand**, which can adapt to different data regimes while avoiding severe distribution shift. We conduct extensive experiments on the downstream image and instruction datasets pruning benchmarks, and the results show that our method establishes *state-of-the-art* performance. Notably, our method significantly speeds up pruning by **35×** in the visual benchmark.

**Limitations** The success of our method relies on the capability of pre-trained models. Thus, our method may fail to improve the downstream performance if the pre-trained model cannot provide meaningful representations.

ACKNOWLEDGMENTS

This research is supported by the Shenzhen Fundamental Research Program (Grant No.JCYJ20230807091809020) and the SUSTech-NUS Joint Research Program. Bingyi Jing is supported in part by the National Natural Science Foundation of China under grant 12371290. We gratefully acknowledge the support of the Center for Computational Science and Engineering at the Southern University of Science and Technology, and the Collaborative Innovation Center of Novel Software Technology and Industrialization at Nanjing University for our research.

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

## A  RELATED WORKS

To accommodate the computation budget, dataset pruning reduces the number of training iterations by selecting the most informative subset. A naive solution evaluates the downstream test performance drop caused by excluding each possible subset, which also underlies the influence function (Ling, 1984; Koh & Liang, 2017; Feldman & Zhang, 2020; Yang et al., 2022) and data Shapley values (Kwon & Zou, 2022; Ghorbani & Zou, 2019). While intuitive, the computation cost is unacceptable because it requires training the model with an additional classification head for $2^n$ **times** given a dataset with size $n$. Therefore, **the inefficiency has cast a key challenge** to identify the critical data subset. Indeed, another line of work turns to replacing the costly counterfactual enumeration with a level-set estimation. Specifically, the most informative samples are identified based on customized scoring functions, which can be roughly divided into the following categories:

- **Geometric:** Herding (Welling, 2009), k-CenterGreedy (Sener & Savarese, 2018), and Contextual Diversity (Agarwal et al., 2020) define the scoring function as the sample similarity in the feature space and redundant points are removed for better diversity. SSP (Sorscher et al., 2022) adopts the k-means algorithm with features from the encoder to search the coreset. Moreover, CCS (Zheng et al., 2023) is an innovative data selection method, which maintains coverage in high-density areas of a dataset.

- **Uncertainty:** Least Confidence, Entropy and Margin (Coleman et al., 2020) select the most uncertain samples that may have a greater impact on model optimization.

- **Error / Loss:** Forgetting (Toneva et al., 2018), GraNd and EL2N (Paul et al., 2021) measure the sample informativeness according to the error or loss during the course of training, and the samples easy to learn are pruned.

- **Decision boundary:** Adversarial DeepFool (Ducoffe & Precioso, 2018) and Contrastive Active Learning (Margatina et al., 2021) preserve samples hard to separate based on the distance to the decision boundary. BoundarySet (Yang et al., 2024) is a novel coreset construction method by selecting training samples to reconstruct the decision boundary of a deep neural network learned on the full dataset.

- **Gradient matching:** CRAIG (Mirzasoleiman et al., 2020) and GRAD-MATCH (Killamsetty et al., 2021) search for a subset with weighted gradients close to the full gradients.

- **Submodularity:** FASS (Wei et al., 2015), PRISM (Kaushal et al., 2021), and SIMILAR (Kothawade et al., 2021) construct the coreset by maximizing the submodular functions (Iyer & Bilmes, 2013), such as Graph Cut, Facility Location and Log Determinant (Iyer et al., 2021), which naturally measures the diversity and information.

To improve the robustness of different dataset pruning methods, the concept of Moderate Coreset (Xia et al., 2023) is discussed. Specifically, given any score criterion of data selection, different scenarios prefer data points with scores in different intervals. However, the above-predefined scoring functions introduce additional pruning costs due to the parameter updating, which is not negligible (Qin et al., 2023). To reduce downstream dataset pruning costs, we propose an efficient scoring function named DLC solely based on the pre-trained model without fine-tuning.

## B  METHOD DETAILS

In this section, we provide a concrete formulation of the pre-training weights masking operation. Given the pre-training weights $W \in \mathbb{R}^{n*m}$ and masking ratio $r \in [0, 1)$, the masking matrix $M \in \{0, 1\}^{n*m}$ is constructed by:

$$M_{i,j} = \begin{cases} 0, & \text{if } |W_{i,j}| < \tau_r \\ 1, & \text{if } |W_{i,j}| \geq \tau_r \end{cases}$$

where $\tau_r$ is the $(n*m*r)$-th element in $\{W_1, ..., W_{n*m}\}$ sorted by L1 norm in ascending order. Finally, the masked pre-training weights $\hat{W}$ can be formulated as:

$$\hat{W} = W \circ M.$$

## C EXPERIMENTS DETAILS

### C.1 SETUPS

#### C.1.1 DOWNSTREAM IMAGE DATASET PRUNING BENCHMARK

Table 5: Downstream datasets for fine-tuning.

| Domain | Downstream Dataset | Training Size | Test Size | Number of Classes | Total Size |
|---|---|---|---|---|---|
| **Medical** | CXRB10 | 4000 | 1000 | 10 | 5000 |
| **Natural** | DeepWeeds | 6400 | 1600 | 8 | 8000 |
| **Texture** | DTD | 3760 | 1880 | 47 | 5640 |
| **Manufacture** | FGVCAircraft | 9000 | 2000 | 100 | 10000 |
| **Illustrative** | Sketch | 16000 | 4000 | 250 | 20000 |

**Baselines.** Aside from the random, we also compare the following methods: Herding (Welling, 2009), kCG (Sener & Savarese, 2018), CD (Agarwal et al., 2020), Forgetting (Toneva et al., 2018), Least Conf, Entropy, Margin (Coleman et al., 2020), GraNd, EL2N[4] (Paul et al., 2021), and SSP (Sorscher et al., 2022). If not specified, one-shot fine-tuning on the full downstream dataset is necessary to estimate the above scoring functions. For the masking-based learning complexity, the size of the learning path increases gradually, starting from the one with initial pre-training weights, until the ranking correlation with the previous no longer changes. The hyperparameter $y$ in different data regimes is selected from the range $\{0.05, 0.10, 0.15, ..., 0.95\}$ by the validation accuracy.

**Fine-tuning.** We sequentially attach a linear layer on top of the pre-trained encoder for the downstream image classification. Then, the above classifier is fully trained on the pruned dataset for 50 epochs using SGD with a momentum of 0.9, a weight decay of 1e-5, and a batch size of 128. The initial learning rate is 1e-3 and decays by a factor of 10 at the 25th and 37th epochs. For a fair comparison, we preserve the model weights at the last epoch for classification evaluation. The above settings are the same for all models and datasets.

**Evaluation** We prune the downstream datasets at 9 pruning ratios, ranging from 10% to 90%, for a thorough verification and comparison. For example, we keep 10% of each category in the original dataset when the pruning ratio is 10%. The efficacy and efficiency of fine-tuning dataset pruning is evaluated by measuring the following metrics: (1) downstream **accuracy** at different pruning ratios. (2) total **time** for scoring estimation, under-sampling, and associated hyperparameter tuning.

**Implementation details.** To ensure reliable reproduction, we have run the compared baselines using the DeepCore (Guo et al., 2022) library[5]. The code is based on PyTorch (Paszke et al., 2019).

#### C.1.2 DOWNSTREAM INSTRUCTION DATASET PRUNING BENCHMARK

**Fine-tuning.** We fine-tune the base model for 3 epochs using SGD with a batch size of 32, a momentum of 0.9, a learning rate of 7e-6 scheduled by cosine function, and a weight decay of 0.01. Note that the learning rate increases linearly at the warmup stage (the first 100 steps). Due to the prohibitive large scale of base pre-trained models, we use the parameter-efficient fine-tuning method LoRA (Hu et al., 2021), which only updates a small number of model parameters for efficiency.

**Evaluation** We prune 50% instructions from the original datasets. For instruction-tuned models, the MMLU with 5 shots is the common benchmark (Hendrycks et al., 2020).

**Implementation details.** Regarding the pre-trained models and instruction datasets, we use the HuggingFace library[6]. Fine-tuning is based on the PEFT library[7], and evaluation is based on the LM-Eval library[8]. The code is based on PyTorch and all the experiments run on NVIDIA L40.

---

[4]We perform three full fine-tuning with different seeds to estimate EL2N score accurately.

[5]https://github.com/PatrickZH/DeepCore

[6]https://huggingface.co

[7]https://github.com/huggingface/peft

[8]https://github.com/EleutherAI/lm-evaluation-harness

## C.2 RESULTS

### C.2.1 MORE RESULTS FOR DLC

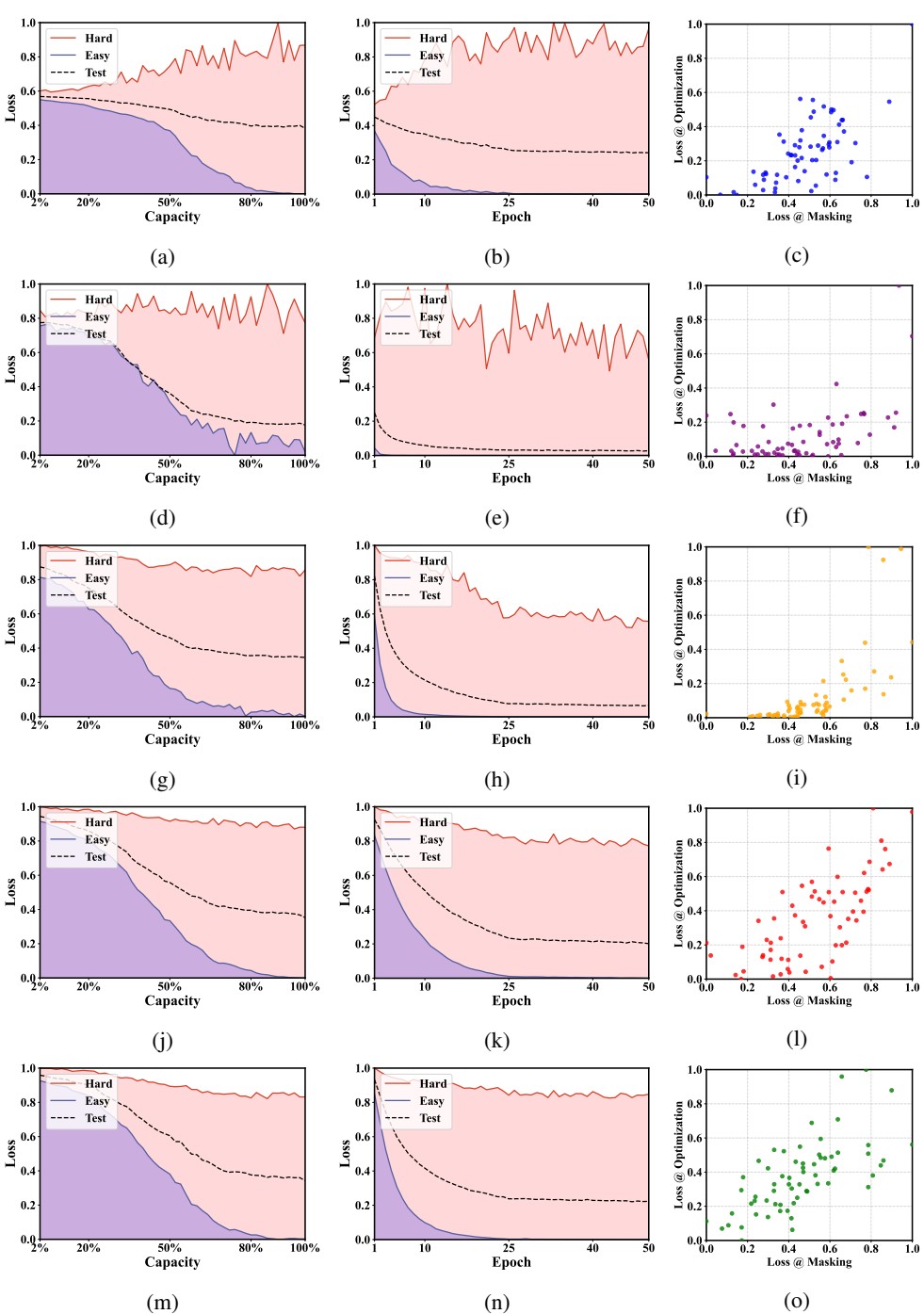

Figure 7: Ranking correlation between the loss integral over the optimization and masking process. The first to fifth rows represent the results on **CXRB10** (Nodule), **DeepWeeds** (Parthenium), **DTD** (Sprinkled), **FGVCAircraft** (DHC-6), and **Sketch** (Revolver) datasets respectively. **(a,d,g,j,m)**: Loss trends with the number of parameters. **(b,e,h,k,n)**: Loss trends with the optimization time. **(c,f,I,l,o)**: High ranking correlation coefficient with $\rho = \{0.54, 0.46, 0.65, 0.68, 0.55\}$.

### C.2.2 DETAILED MAIN RESULTS

Table 6: Classification accuracy (%) over **5** diverse downstream datasets and pruning 3 different average ratios with the fully pre-trained ResNet-18.

| Method | CXRB10 | | | DeepWeeds | | | DTD | | | FGVCAircraft | | | Sketch | | | Average |
|---|---|---|---|---|---|---|---|---|---|---|---|---|---|---|---|---|
| | 20% | 50% | 80% | 20% | 50% | 80% | 20% | 50% | 80% | 20% | 50% | 80% | 20% | 50% | 80% | |
| Random | 24.90 | 31.13 | 33.60 | 83.83 | 91.71 | 93.42 | 51.83 | 63.44 | 67.75 | 20.87 | 42.75 | 55.44 | 41.19 | 63.88 | 70.82 | 55.77 |
| Herding | 24.73 | 31.20 | 33.87 | 48.50 | 76.65 | 91.17 | 31.38 | 54.43 | 66.38 | 13.85 | 33.36 | 52.76 | 29.16 | 57.59 | 69.40 | 47.63 |
| kCG | 23.63 | 29.73 | 34.00 | 82.58 | 91.71 | 93.75 | 46.99 | 64.20 | 68.19 | 15.45 | 37.72 | 53.80 | 35.00 | 63.06 | 71.13 | 54.06 |
| CD | 20.77 | 29.33 | 34.33 | 81.88 | 93.02 | 94.21 | 43.65 | 63.95 | 68.48 | 15.78 | 36.73 | 54.08 | 33.03 | 61.68 | 70.69 | 53.44 |
| Least Conf | 27.10 | 32.20 | 34.70 | 77.98 | 87.27 | 91.54 | 52.85 | 61.40 | 66.76 | 25.94 | 46.13 | 56.37 | 48.43 | 64.61 | 70.43 | 56.25 |
| Entropy | 27.53 | 32.37 | 33.93 | 77.94 | 87.79 | 91.94 | 52.43 | 61.79 | 66.63 | 26.12 | 46.06 | 56.58 | 48.55 | 64.88 | 70.63 | 56.35 |
| Margin | 26.30 | 32.67 | 34.10 | 77.73 | 87.75 | 92.02 | 52.93 | 61.26 | 67.04 | 25.71 | 45.79 | 56.18 | 48.56 | 64.63 | 70.63 | 56.22 |
| GraNd | 19.50 | 30.73 | 34.27 | 80.48 | 92.67 | 94.29 | 40.05 | 62.66 | 68.53 | 15.05 | 35.52 | 54.36 | 31.14 | 59.97 | 70.93 | 52.68 |
| EL2N | 14.80 | 28.67 | 34.27 | 80.54 | 92.75 | 94.08 | 38.35 | 62.61 | 68.85 | 14.94 | 35.71 | 54.29 | 29.98 | 59.71 | 71.25 | 52.05 |
| Forgetting | 25.80 | 33.10 | 35.63 | 81.33 | 89.94 | 93.10 | 50.39 | 64.20 | 68.26 | 19.13 | 39.90 | 55.11 | 40.74 | 63.08 | 71.03 | 55.38 |
| Ours | 27.23 | 32.87 | 35.33 | 85.21 | 92.19 | 93.58 | 53.67 | 65.39 | 68.21 | 22.28 | 43.61 | 56.02 | 44.37 | 65.05 | 71.15 | 57.08 |

Table 7: Classification accuracy (%) over **5** diverse downstream datasets and pruning 3 different average ratios with the fully pre-trained ResNet-50.

| Method | CXRB10 | | | DeepWeeds | | | DTD | | | FGVCAircraft | | | Sketch | | | Average |
|---|---|---|---|---|---|---|---|---|---|---|---|---|---|---|---|---|
| | 20% | 50% | 80% | 20% | 50% | 80% | 20% | 50% | 80% | 20% | 50% | 80% | 20% | 50% | 80% | |
| Random | 26.30 | 31.67 | 36.37 | 86.50 | 93.75 | 94.85 | 59.95 | 70.43 | 73.44 | 24.78 | 51.16 | 63.81 | 45.66 | 69.01 | 75.26 | 60.19 |
| Herding | 24.80 | 32.47 | 36.00 | 53.75 | 83.81 | 94.29 | 35.78 | 59.82 | 70.62 | 17.90 | 42.53 | 61.37 | 35.52 | 62.63 | 73.78 | 52.34 |
| kCG | 23.80 | 30.70 | 35.37 | 86.50 | 94.50 | 95.58 | 52.66 | 70.00 | 74.10 | 19.07 | 45.26 | 62.88 | 39.88 | 67.79 | 75.47 | 58.24 |
| CD | 22.30 | 31.27 | 35.13 | 84.75 | 94.50 | 95.67 | 53.81 | 69.86 | 74.77 | 20.37 | 45.81 | 63.37 | 39.63 | 66.89 | 75.24 | 58.22 |
| Least Conf | 26.93 | 32.63 | 36.03 | 83.02 | 90.31 | 93.94 | 60.82 | 68.74 | 72.64 | 28.47 | 52.33 | 64.74 | 52.01 | 68.23 | 74.52 | 60.36 |
| Entropy | 26.70 | 32.13 | 35.40 | 83.52 | 90.40 | 94.10 | 60.96 | 68.97 | 72.57 | 28.78 | 53.02 | 64.81 | 52.27 | 68.54 | 74.47 | 60.44 |
| Margin | 25.60 | 33.37 | 35.87 | 83.23 | 90.10 | 94.25 | 60.14 | 68.71 | 72.13 | 28.26 | 51.73 | 64.42 | 51.13 | 68.63 | 74.53 | 60.14 |
| GraNd | 20.10 | 31.63 | 36.90 | 82.98 | 94.40 | 96.04 | 51.45 | 68.76 | 74.06 | 20.10 | 45.09 | 63.59 | 38.89 | 65.98 | 74.93 | 57.66 |
| EL2N | 17.70 | 32.40 | 36.60 | 83.00 | 94.48 | 95.98 | 49.68 | 69.33 | 74.36 | 19.60 | 44.63 | 63.38 | 36.73 | 65.27 | 74.98 | 57.21 |
| Forgetting | 27.13 | 32.70 | 37.00 | 84.08 | 92.00 | 94.98 | 57.98 | 71.44 | 74.18 | 23.48 | 48.92 | 63.60 | 46.78 | 68.57 | 74.78 | 59.84 |
| Ours | 29.83 | 34.63 | 37.10 | 87.77 | 94.06 | 95.69 | 61.58 | 70.69 | 74.73 | 26.72 | 51.59 | 64.56 | 48.85 | 69.83 | 75.60 | 61.55 |

Table 8: Classification accuracy (%) over **5** diverse downstream datasets and pruning 3 different average ratios with the fully pre-trained ViT-Small.

| Method | CXRB10 | | | DeepWeeds | | | DTD | | | FGVCAircraft | | | Sketch | | | Average |
|---|---|---|---|---|---|---|---|---|---|---|---|---|---|---|---|---|
| | 20% | 50% | 80% | 20% | 50% | 80% | 20% | 50% | 80% | 20% | 50% | 80% | 20% | 50% | 80% | |
| Random | 27.93 | 33.20 | 36.73 | 88.60 | 94.40 | 95.56 | 58.28 | 68.63 | 73.62 | 28.61 | 52.00 | 63.68 | 51.46 | 71.64 | 76.59 | 61.40 |
| Herding | 25.93 | 31.70 | 37.13 | 80.54 | 91.33 | 95.04 | 54.56 | 67.32 | 72.82 | 25.55 | 47.20 | 63.16 | 49.86 | 71.08 | 76.24 | 59.30 |
| kCG | 27.33 | 33.90 | 38.60 | 88.02 | 93.46 | 95.77 | 57.41 | 69.63 | 73.81 | 30.08 | 52.37 | 63.34 | 52.71 | 71.62 | 76.59 | 61.64 |
| CD | 23.97 | 32.87 | 37.03 | 89.31 | 94.81 | 95.98 | 54.04 | 70.25 | 73.99 | 27.20 | 52.94 | 64.04 | 47.78 | 71.92 | 76.98 | 60.87 |
| Least Conf | 30.37 | 35.27 | 37.07 | 80.02 | 90.04 | 94.40 | 59.49 | 68.53 | 72.02 | 29.91 | 48.86 | 61.59 | 54.20 | 69.44 | 76.08 | 60.49 |
| Entropy | 30.37 | 35.17 | 37.40 | 80.50 | 90.69 | 94.60 | 58.83 | 68.87 | 72.00 | 30.70 | 49.38 | 61.54 | 54.27 | 69.86 | 75.93 | 60.67 |
| Margin | 30.93 | 34.67 | 38.00 | 80.46 | 90.65 | 94.56 | 59.41 | 68.17 | 72.46 | 29.30 | 48.65 | 61.26 | 54.24 | 70.13 | 75.98 | 60.59 |
| GraNd | 22.73 | 32.07 | 37.27 | 89.54 | 94.94 | 95.81 | 53.67 | 69.34 | 74.31 | 22.30 | 54.09 | 64.26 | 46.19 | 71.20 | 76.86 | 60.70 |
| EL2N | 21.47 | 32.57 | 37.13 | 88.60 | 95.10 | 96.04 | 52.00 | 68.72 | 73.65 | 27.60 | 53.46 | 64.35 | 46.18 | 71.39 | 77.08 | 60.36 |
| Forgetting | 30.40 | 34.93 | 36.13 | 87.35 | 93.25 | 95.06 | 58.00 | 71.40 | 74.73 | 31.52 | 54.60 | 64.35 | 52.88 | 69.99 | 76.48 | 62.07 |
| Ours | 31.33 | 36.50 | 38.70 | 89.88 | 94.56 | 95.81 | 60.35 | 70.87 | 73.92 | 29.40 | 52.98 | 63.85 | 53.72 | 72.03 | 76.93 | 62.72 |

Table 9: Classification accuracy (%) over **5** diverse downstream datasets and pruning 3 different average ratios with the fully pre-trained ViT-Base.

| Method | CXRB10 | | | DeepWeeds | | | DTD | | | FGVCAircraft | | | Sketch | | | Average |
|---|---|---|---|---|---|---|---|---|---|---|---|---|---|---|---|---|
| | 20% | 50% | 80% | 20% | 50% | 80% | 20% | 50% | 80% | 20% | 50% | 80% | 20% | 50% | 80% | |
| Random | 26.20 | 32.07 | 34.30 | 86.50 | 93.15 | 94.71 | 55.32 | 67.43 | 70.25 | 26.38 | 49.72 | 61.22 | 45.03 | 69.28 | 75.46 | 59.13 |
| Herding | 24.33 | 32.40 | 35.67 | 82.46 | 90.94 | 93.77 | 49.49 | 64.41 | 70.51 | 20.97 | 44.17 | 60.41 | 39.68 | 67.61 | 75.07 | 56.79 |
| kCG | 28.23 | 31.77 | 35.23 | 87.00 | 93.02 | 94.58 | 55.20 | 67.11 | 70.43 | 25.05 | 48.94 | 61.03 | 41.86 | 69.83 | 76.03 | 59.02 |
| CD | 22.17 | 31.80 | 35.77 | 85.90 | 94.48 | 95.10 | 46.84 | 66.33 | 71.17 | 21.16 | 46.24 | 61.19 | 37.01 | 68.18 | 76.25 | 57.31 |
| Least Conf | 28.70 | 33.73 | 35.07 | 78.44 | 89.13 | 93.25 | 56.10 | 64.96 | 69.56 | 30.70 | 50.16 | 60.37 | 52.60 | 69.11 | 75.13 | 59.13 |
| Entropy | 29.23 | 33.87 | 35.93 | 78.23 | 88.73 | 93.52 | 56.10 | 64.70 | 69.89 | 30.90 | 50.57 | 60.72 | 53.16 | 69.26 | 75.03 | 59.32 |
| Margin | 29.40 | 33.47 | 36.60 | 78.52 | 88.96 | 93.60 | 56.51 | 65.12 | 69.89 | 29.95 | 50.31 | 60.50 | 53.17 | 68.97 | 75.46 | 59.36 |
| GraNd | 22.07 | 31.83 | 35.60 | 85.48 | 94.38 | 94.81 | 44.08 | 66.26 | 71.28 | 18.82 | 46.23 | 62.15 | 36.24 | 67.37 | 75.68 | 56.82 |
| EL2N | 19.03 | 29.87 | 34.23 | 84.77 | 94.60 | 95.02 | 41.79 | 65.80 | 70.85 | 20.23 | 45.53 | 61.86 | 33.47 | 66.80 | 76.13 | 56.00 |
| Forgetting | 28.33 | 33.77 | 36.47 | 85.90 | 92.13 | 94.46 | 54.33 | 68.46 | 71.35 | 24.78 | 49.21 | 61.85 | 48.75 | 69.33 | 75.18 | 59.62 |
| Ours | 28.93 | 34.77 | 36.27 | 88.35 | 93.44 | 94.79 | 56.77 | 67.85 | 71.01 | 27.60 | 50.96 | 61.93 | 50.25 | 71.08 | 76.46 | 60.70 |

### C.2.3 More Results for Weakly Pre-trained Models

Table 10: Classification accuracy (%) over **5** diverse downstream datasets and pruning 3 different average ratios with the weakly pre-trained ResNet-18.

| Method | CXRB10 | | | DeepWeeds | | | DTD | | | FGVCAircraft | | | Sketch | | | Average |
|---|---|---|---|---|---|---|---|---|---|---|---|---|---|---|---|---|
| | 20% | 50% | 80% | 20% | 50% | 80% | 20% | 50% | 80% | 20% | 50% | 80% | 20% | 50% | 80% | |
| Random | 26.17 | 32.43 | 35.07 | 87.04 | 93.15 | 94.52 | 54.86 | 66.19 | 69.29 | 32.37 | 55.89 | 66.65 | 56.01 | 72.90 | 77.13 | 61.31 |
| Herding | 23.60 | 30.57 | 35.37 | 51.60 | 78.96 | 90.73 | 31.99 | 56.13 | 66.81 | 24.78 | 50.96 | 66.33 | 44.05 | 67.25 | 76.35 | 53.03 |
| kCG | 24.27 | 30.33 | 35.57 | 87.42 | 93.81 | 94.75 | 49.10 | 65.18 | 70.12 | 29.06 | 55.44 | 67.40 | 54.98 | 73.11 | 77.22 | 60.52 |
| CD | 21.87 | 30.33 | 35.70 | 87.15 | 94.15 | 94.96 | 53.23 | 66.21 | 70.96 | 35.36 | 59.82 | 68.59 | 54.17 | 73.25 | 78.09 | 61.59 |
| Least Conf | 27.07 | 31.87 | 35.67 | 83.75 | 90.02 | 93.46 | 56.08 | 64.17 | 68.79 | 30.78 | 51.61 | 64.19 | 58.12 | 71.19 | 76.52 | 60.22 |
| Entropy | 28.50 | 32.20 | 35.13 | 83.13 | 90.19 | 93.77 | 55.85 | 64.10 | 68.76 | 30.99 | 51.64 | 64.44 | 58.45 | 71.14 | 76.59 | 60.32 |
| Margin | 28.70 | 33.27 | 36.00 | 83.17 | 90.04 | 93.58 | 56.26 | 63.71 | 67.85 | 30.29 | 50.85 | 64.55 | 58.29 | 71.44 | 76.46 | 60.30 |
| GraNd | 21.33 | 30.80 | 35.33 | 86.00 | 94.06 | 94.77 | 54.22 | 66.54 | 70.11 | 36.14 | 60.41 | 68.75 | 54.16 | 73.03 | 77.80 | 61.56 |
| EL2N | 18.63 | 30.37 | 35.40 | 87.63 | 94.54 | 95.00 | 53.51 | 66.58 | 70.21 | 35.56 | 61.23 | 69.07 | 53.07 | 72.81 | 77.33 | 61.40 |
| Forgetting | 29.10 | 34.60 | 35.63 | 86.85 | 92.31 | 94.54 | 54.02 | 67.18 | 70.62 | 36.56 | 60.58 | 68.05 | 58.85 | 72.93 | 76.73 | 62.57 |
| Ours | 29.90 | 34.33 | 36.83 | 88.50 | 93.46 | 95.00 | 57.48 | 66.67 | 70.37 | 33.38 | 57.63 | 67.32 | 58.07 | 73.14 | 77.37 | 62.63 |

Table 11: Classification accuracy (%) over **5** diverse downstream datasets and pruning 3 different average ratios with the weakly pre-trained ResNet-50.

| Method | CXRB10 | | | DeepWeeds | | | DTD | | | FGVCAircraft | | | Sketch | | | Average |
|---|---|---|---|---|---|---|---|---|---|---|---|---|---|---|---|---|
| | 20% | 50% | 80% | 20% | 50% | 80% | 20% | 50% | 80% | 20% | 50% | 80% | 20% | 50% | 80% | |
| Random | 27.63 | 34.37 | 37.23 | 90.31 | 95.33 | 96.44 | 63.21 | 72.75 | 75.53 | 42.55 | 66.65 | 76.64 | 66.80 | 78.62 | 82.13 | 67.08 |
| Herding | 21.03 | 31.33 | 37.03 | 53.96 | 86.85 | 95.63 | 39.26 | 63.26 | 74.08 | 34.58 | 62.00 | 75.14 | 54.66 | 74.17 | 80.72 | 58.91 |
| kCG | 25.97 | 33.17 | 37.53 | 90.63 | 96.17 | 97.02 | 57.15 | 72.36 | 76.42 | 40.02 | 67.19 | 77.11 | 66.24 | 79.77 | 82.33 | 66.60 |
| CD | 24.23 | 32.50 | 34.80 | 91.40 | 96.15 | 97.04 | 60.94 | 72.66 | 76.60 | 46.56 | 69.98 | 78.06 | 66.09 | 79.13 | 82.43 | 67.24 |
| Least Conf | 28.47 | 35.70 | 37.80 | 87.73 | 92.92 | 96.25 | 60.83 | 70.53 | 75.00 | 41.63 | 63.31 | 74.36 | 67.45 | 77.27 | 81.62 | 66.06 |
| Entropy | 27.97 | 34.47 | 36.87 | 87.19 | 93.48 | 96.06 | 61.06 | 70.46 | 75.14 | 41.82 | 63.56 | 75.01 | 67.52 | 77.35 | 81.63 | 65.97 |
| Margin | 29.23 | 34.33 | 37.00 | 87.40 | 93.48 | 96.15 | 61.15 | 70.23 | 75.14 | 41.34 | 63.25 | 74.80 | 67.40 | 77.43 | 81.67 | 66.00 |
| GraNd | 24.30 | 32.63 | 37.03 | 91.19 | 96.46 | 96.88 | 60.87 | 73.51 | 76.47 | 47.36 | 70.53 | 77.75 | 65.63 | 79.15 | 82.15 | 67.46 |
| EL2N | 24.37 | 33.10 | 37.60 | 90.98 | 96.35 | 96.56 | 60.34 | 72.93 | 76.19 | 47.47 | 70.36 | 78.72 | 64.62 | 79.27 | 82.27 | 67.41 |
| Forgetting | 29.33 | 35.43 | 36.70 | 90.02 | 94.08 | 96.63 | 62.22 | 73.28 | 76.21 | 46.72 | 70.48 | 77.46 | 67.10 | 78.88 | 82.03 | 67.77 |
| Ours | 30.40 | 35.73 | 37.87 | 91.90 | 95.96 | 96.88 | 64.57 | 73.55 | 76.37 | 43.66 | 68.29 | 77.23 | 67.85 | 79.53 | 82.36 | 68.14 |

Table 12: Classification accuracy (%) over **5** diverse downstream datasets and pruning 3 different average ratios with the weakly pre-trained ViT-Small.

| Method | CXRB10 | | | DeepWeeds | | | DTD | | | FGVCAircraft | | | Sketch | | | Average |
|---|---|---|---|---|---|---|---|---|---|---|---|---|---|---|---|---|
| | 20% | 50% | 80% | 20% | 50% | 80% | 20% | 50% | 80% | 20% | 50% | 80% | 20% | 50% | 80% | |
| Random | 28.50 | 33.80 | 37.07 | 91.31 | 95.75 | 96.73 | 60.71 | 73.63 | 76.67 | 31.87 | 56.81 | 68.46 | 54.56 | 74.08 | 79.14 | 63.94 |
| Herding | 28.17 | 33.77 | 36.30 | 81.98 | 94.71 | 96.60 | 54.72 | 71.47 | 76.47 | 27.14 | 54.16 | 67.56 | 50.98 | 73.31 | 79.07 | 61.76 |
| kCG | 25.93 | 33.77 | 36.63 | 92.19 | 96.19 | 96.98 | 63.28 | 74.38 | 77.80 | 33.60 | 57.92 | 67.70 | 54.56 | 74.72 | 79.33 | 64.33 |
| CD | 21.80 | 33.00 | 37.43 | 91.17 | 96.67 | 97.40 | 56.28 | 73.81 | 77.70 | 33.64 | 59.72 | 68.76 | 49.98 | 75.74 | 80.19 | 63.55 |
| Least Conf | 27.77 | 33.30 | 36.30 | 86.21 | 92.29 | 96.08 | 61.79 | 71.70 | 76.06 | 31.76 | 52.78 | 66.18 | 56.90 | 72.39 | 78.55 | 62.67 |
| Entropy | 29.80 | 34.97 | 35.60 | 86.10 | 92.67 | 95.98 | 62.04 | 71.56 | 75.05 | 30.78 | 53.20 | 65.98 | 57.19 | 72.04 | 78.48 | 62.76 |
| Margin | 28.70 | 33.93 | 37.00 | 85.96 | 92.65 | 95.92 | 62.30 | 72.23 | 75.85 | 31.95 | 53.44 | 65.97 | 56.95 | 72.43 | 78.27 | 62.90 |
| GraNd | 20.17 | 31.67 | 36.23 | 91.44 | 96.54 | 97.10 | 54.15 | 73.74 | 77.59 | 34.85 | 59.70 | 69.28 | 49.18 | 74.66 | 79.99 | 63.09 |
| EL2N | 19.97 | 32.73 | 36.00 | 90.90 | 97.25 | 97.42 | 52.04 | 73.23 | 77.43 | 33.30 | 59.82 | 69.84 | 48.58 | 74.83 | 79.83 | 62.88 |
| Forgetting | 28.53 | 33.03 | 36.40 | 88.88 | 94.96 | 97.06 | 64.18 | 75.37 | 77.59 | 35.93 | 59.80 | 69.03 | 56.76 | 73.06 | 78.88 | 64.63 |
| Ours | 30.03 | 35.90 | 37.53 | 92.48 | 96.40 | 97.38 | 64.79 | 74.72 | 77.50 | 33.72 | 58.15 | 69.06 | 56.12 | 74.80 | 79.58 | 65.21 |

Table 13: Classification accuracy (%) over **5** diverse downstream datasets and pruning 3 different average ratios with the weakly pre-trained ViT-Base.

| Method | CXRB10 | | | DeepWeeds | | | DTD | | | FGVCAircraft | | | Sketch | | | Average |
|---|---|---|---|---|---|---|---|---|---|---|---|---|---|---|---|---|
| | 20% | 50% | 80% | 20% | 50% | 80% | 20% | 50% | 80% | 20% | 50% | 80% | 20% | 50% | 80% | |
| Random | 20.40 | 31.07 | 32.37 | 93.92 | 97.52 | 97.81 | 58.49 | 74.06 | 78.53 | 37.57 | 62.21 | 72.33 | 55.26 | 78.52 | 82.33 | 64.83 |
| Herding | 17.03 | 25.50 | 29.33 | 84.48 | 96.50 | 97.48 | 55.30 | 71.45 | 78.30 | 32.06 | 60.41 | 71.09 | 50.78 | 77.32 | 81.93 | 61.93 |
| kCG | 19.70 | 28.23 | 34.73 | 93.17 | 97.08 | 97.88 | 61.33 | 72.93 | 78.26 | 37.66 | 63.36 | 72.46 | 62.05 | 79.13 | 82.16 | 65.34 |
| CD | 17.70 | 27.07 | 30.90 | 72.31 | 97.75 | 98.04 | 58.10 | 73.01 | 78.78 | 37.40 | 63.86 | 73.79 | 56.26 | 64.69 | 82.82 | 62.17 |
| Least Conf | 18.70 | 29.57 | 32.67 | 86.10 | 93.65 | 97.00 | 60.66 | 71.74 | 76.70 | 31.50 | 58.37 | 70.73 | 62.93 | 78.08 | 81.98 | 63.36 |
| Entropy | 18.10 | 30.83 | 33.90 | 84.65 | 94.02 | 97.10 | 61.90 | 71.86 | 76.99 | 35.79 | 59.62 | 71.19 | 62.38 | 77.68 | 81.76 | 63.85 |
| Margin | 20.07 | 29.50 | 29.13 | 84.94 | 93.96 | 96.83 | 63.49 | 70.60 | 75.14 | 36.11 | 58.28 | 70.70 | 62.48 | 76.87 | 82.02 | 63.34 |
| GraNd | 16.07 | 27.97 | 34.57 | 93.77 | 97.31 | 98.21 | 54.06 | 73.60 | 78.67 | 38.08 | 64.83 | 72.85 | 56.11 | 78.67 | 60.98 | 63.05 |
| EL2N | 15.00 | 26.60 | 33.13 | 94.23 | 97.58 | 98.02 | 56.95 | 72.06 | 78.88 | 39.54 | 65.51 | 73.59 | 57.55 | 79.07 | 82.84 | 64.70 |
| Forgetting | 21.43 | 25.83 | 31.93 | 92.21 | 96.23 | 97.96 | 60.20 | 75.35 | 78.39 | 40.61 | 64.62 | 73.09 | 59.76 | 78.56 | 82.34 | 65.23 |
| Ours | 23.87 | 31.57 | 33.93 | 94.77 | 97.67 | 98.08 | 65.69 | 75.35 | 79.36 | 38.95 | 63.92 | 72.59 | 62.26 | 79.45 | 82.52 | 66.67 |

### C.2.4  COMPARISON OF DIFFERENT UNDER-SAMPLING STRATEGIES

Table 14: Average classification accuracy (%) over 9 pruning ratios with the fully pre-trained ResNet-18 from different under-sampling strategies. **Bold** numbers are the optimal results.

| Under-sampling Strategy | CXRB10 | DeepWeeds | DTD | FGVCAircraft | Sketch | Average |
|---|---|---|---|---|---|---|
| **Random** | 29.88 | 89.65 | 61.01 | 39.69 | 58.63 | 55.77 |
| **DQ** | 30.07 | 89.49 | 61.17 | 40.34 | 58.82 | 55.98 |
| **FlexRand** | **31.81** | **90.33** | **62.42** | **40.64** | **60.19** | **57.08** |

In this section, we further compare different under-sampling strategies to verify the superiority of FlexRand. Specifically, the strategy in Dataset Quantization (Zhou et al., 2023b) dividing datasets into different bins and randomly sampling from each bin, alleviates distribution shift and is similar to ours. Differently, FlexRand divides the full dataset into two bins with different sizes: easy and hard bins. In this way, samples in different bins have different probabilities of being selected, which enables adaptation in different data regimes. Moreover, we empirically compare the performance of FlexRand and Dataset Quantization (DQ) using the same DLC score. The experiments are conducted with the fully pre-trained ResNet-18 model and we keep the same fine-tuning setting as in the manuscript. As shown in Table 14, FlexRand outperforms the DQ, verifying its superiority.

### C.2.5  PERFORMANCE OF PRE-TRAINED MODELS BY LINEAR PROBING

Table 15: Accuracy (%) of pre-trained models by linear probing on the full downstream dataset.

| Dataset | Weakly Supervised | | | | Fully Supervised | | | |
|---|---|---|---|---|---|---|---|---|
| | RN18 | RN50 | ViT-Small | ViT-Base | RN18 | RN50 | ViT-Small | ViT-Base |
| **CXRB10** | 26.10 | 30.70 | 38.70 | 35.10 | 25.40 | 27.10 | 35.80 | 36.70 |
| **DeepWeeds** | 80.88 | 87.38 | 97.06 | 98.13 | 77.38 | 80.50 | 95.94 | 94.31 |
| **DTD** | 67.87 | 74.95 | 77.61 | 78.51 | 61.70 | 66.12 | 75.69 | 71.76 |
| **FGVCAircraft** | 35.55 | 46.38 | 69.10 | 71.92 | 27.96 | 30.06 | 64.21 | 61.12 |
| **Sketch** | 58.00 | 63.05 | 79.70 | 81.95 | 50.28 | 51.43 | 76.50 | 74.93 |

As shown in Table 15, we present the classification accuracy of those pre-trained models by linear probing by linear probing on the full downstream dataset.

