# OpenReview forum: "Exploring Learning Complexity for Efficient Downstream Dataset Pruning"
_ICLR.cc/2025/Conference — ICLR 2025 Poster_

### Official Review · Reviewer_aZeo · 2024-10-28

**Soundness:** 3
**Presentation:** 3
**Contribution:** 3
**Rating:** 8
**Confidence:** 4

**Summary:**

This work describes a novel dataset pruning method without the need of pre-training on the target dataset. Given models pre-trained on large scale datasets, this work proposes a Distorting-based Learning Complexity score to identify informative images and instructions. Sample hardness is estimated by randomly masked neural networks, representing networks with different capabilities. Then samples are randomly sampled from the easy and hard groups, respectively. The proposed method achieves effective dataset pruning with 35x less pruning time.

**Strengths:**

1. The design of using random masks to produce classifiers with different capabilities is interesting and practical. With the averaged feature serving as the prediction head, there is no more need to fine-tune the classifier on downstream tasks.
2. Detailed experiments are conducted to illustrate the effetiveness of the proposed method. The method can be applied to both image and instruction datasets, both demonstrating performance improvement.
3. The writing is generally fluent and easy to follow.

**Weaknesses:**

1. The authors claim that easy samples are more likely to be correctly classified by a weak classifier in the front part of the learning path. However, the overall Learning Complexity score is acquired by averaging classification loss of multiple randomly sampled networks. The definition of learning path seems not to be utilized in the method design.
2. Can the utilization also be applied to some of previous methods? For example, the Herding method uses parameter influence as scores for each sample. Here the fine-tuned model can also be substituted by a pre-trained model with averaged features as the prediction head. Although the direct employment of pre-trained models is practical, it is not a unique design. And it will be interesting to see if applying the strategy to previous methods also leads to performance improvement.
3. The strategy of dividing datasets into different groups and randomly sampling from each group is similar to the idea in Dataset Quantization [1]. Dataset Quantization first iteratively separate the data into multiple bins with coreset selection methods. Normally the early groups tend to cluster around the distribution center, while the later groups show more diversity. By sampling from each bin, the overall distribution will be kept similar to the original one. This paper has a similar claim that FlexRand avoids severe distribution shift. Please discuss the difference of the proposed strategy from Dataset Quantization and the advantages of it.
4. Section 5 discusses the quality of pre-trained models. The authors claim that the method is not sensitive to the quality of pre-trained models. But weakly supervised models are not always worse than fully supervised models. Please also show the original performance comparison between these two groups of models.
5. Minor:
    - The use of pretrain and pre-train need to be unified in the paper.
    - The sample number is represented both by N (line 105) and |D| (line 129). Please unify the use.

[1] Zhou, Daquan, et al. "Dataset quantization." Proceedings of the IEEE/CVF International Conference on Computer Vision. 2023.

**Questions:**

1. How is the loss integration implemented? In the integration figures, the upper bound of loss is 1.0. Is it normalized to the range of (0, 1)?
2. How is the masking applied to the neural network?
3. How is the splitting hyper-parameter $\gamma$ determined in the actual use? If multiple values need to be tested, the tuning time should also be counted towards the pruning time in Figure 1.

---

> ### Author Response · Authors · 2024-11-19
> **Response to Reviewer aZeo (1/2)**
>
> We appreciate the reviewer for the insightful and detailed comments. Please find our response below:
>
> ### **1. Role of the Learning Path [W1]**
> Thank you for pointing out the potential for misunderstanding. We first clarify that subnets with different numbers of parameters are deterministically produced with L1-based masking operation (line 197-199) instead of randomly. Such subnets with different capabilities constitute a viable learning path and allow us to distinguish between easy and hard samples as shown in Figure 2(a).
>
> Importantly, we calculate the Learning Complexity score by approximating the definite integral of loss in the learning path. For efficiency, we sum the classification losses from subnets in the above learning path. Therefore, our method is designed to implement the learning path in a computationally efficient manner, which enables broader applications in practice.
>
>
> ### **2. Employment of pre-trained models for previous methods [W2]**
> Yes, the strategy can be directly applied to several previous methods. Here, we compare the performance of the original dataset pruning (**FT**) with the variants of pre-training models with average features as prediction head (**PT**), on several methods including Herding [2], k-CenterGreedy (kCG) [3] and Contextual Diversity (CD) [4]. The experiments are conducted with the fully pre-trained ResNet-18 model and we keep the same fine-tuning setting as in the manuscript.
>
> As shown in the Table below, using our strategy can improve the average classification performance of these three pruning algorithms, over 5 downstream datasets and 9 pruning ratios. For example, the variant of CD outperforms the vanilla CD by **1.63**\% on average. However, those methods still perform worse than the random strategy, while our method obtains the best performance.
>
> |              | CXRB10   | DeepWeeds | DTD      | FGVCAircraft | Sketch   | Average  |
> | -------      | -------- | --------- | -------- | ------------ | -------- | -------- |
> | Random       | 29.88    | 89.65     | 61.01    | 39.69        | 58.63    | 55.77    |
> | Herding (PT) | 26.62    | 76.14     | 51.45    | 33.03        | 53.14    | 48.08    |
> | Herding (FT) | 29.93    | 72.11     | 50.73    | 33.32        | 52.05    | 47.63    |
> | kCG (PT)     | 29.68    | 88.85     | 59.61    | 38.80        | 57.16    | 54.82    |
> | kCG (FT)     | 29.12    | 89.35     | 59.79    | 35.66        | 56.40    | 54.06    |
> | CD (PT)      | 29.48    | 89.40     | 59.92    | 39.26        | 57.28    | 55.07    |
> | CD (FT)      | 28.14    | 89.70     | 58.69    | 35.53        | 55.13    | 53.44    |
> | Ours         | **31.81**| **90.33** | **62.42**| **40.64**    | **60.19**| **57.08**|
>
>
>
> ### **3. Comparison of FlexRand and the strategy from Dataset Quantization [W3]**
> Yes, the proposed strategy for alleviating distribution shift is similar but different from the idea in Dataset Quantization [1]. Specifically, the strategy in Dataset Quantization divides the full dataset into **multiple bins of the same size**, and the number of bins is defined as a hyperparameter. Differently, FlexRand divides the full dataset into **two bins with different sizes: easy and hard bins**, and we use the splitting hyper-parameter $\gamma\in(0,1)$ to split the dataset. In this way, samples in different bins have different probabilities of being selected, which enables adaptation in different settings. As recognized by Reviewer N2di, this FlexRand *makes it robust across different data regimes*, which is *a valuable contribution to data pruning strategies*.
>
> Moreover, we empirically compare the performance of FlexRand and Dataset Quantization (DQ) using the same DLC score. The experiments are conducted with the fully pre-trained ResNet-18 model and we keep the same fine-tuning setting as in the manuscript. For the strategy in Dataset Quantization, we search the hyper-parameter within the default range {1, 5, 10, 20}, with the same method as the splitting $\gamma$ (see the 8th answer about hyper-parameter tuning). As shown in the Table below, FlexRand outperforms the strategy in DQ with better average classification performance over 5 downstream datasets and 9 pruning ratios. We add this comparison of different under-sampling strategies to Appendix C.2.4 of the manuscript.
>
>
> | Strategy | CXRB10   | DeepWeeds | DTD      | FGVCAircraft | Sketch   | Average  |
> | -------- | -------- | --------- | -------- | ------------ | -------- | -------- |
> | Random   | 29.88    | 89.65     | 61.01    | 39.69        | 58.63    | 55.77    |
> | DQ       | 30.07    | 89.49     | 61.17    | 40.34        | 58.82    | 55.98    |
> | FlexRand | **31.81**| **90.33** | **62.42**| **40.64**    | **60.19**| **57.08**|

---

> > ### Author Response · Authors · 2024-11-19
> > **Response to Reviewer aZeo (2/2)**
> >
> > ### **4. Sensitivity analysis on the quality of pre-trained models [W4]**
> > Thank you for pointing out the mistake in the discussion. We clarify that the analysis is to investigate the quality of pre-training data, instead of the model quality. To avoid any misunderstanding, we fix the description in the Discussion section. As suggested by the reviewer, we present in Appendix C.2.5 the classification accuracy of pre-trained models, by linear probing on the full downstream dataset. It is true that weakly supervised models are not always worse than fully supervised models.
> >
> >
> > ### **5. Typos [W5]**
> > Thank you for pointing typos out. We have fixed these in the updated version.
> >
> >
> > ### **6. Detail about loss integration [Q1]**
> > For computational efficiency, we sample multiple masked rates to approximate the definite integral of loss. In practice, we implement this by accumulating the classification losses from subnets with various masked rates (line 199-204). For the integral figures, we utilize Max-Min normalization to scale the loss to the range of (0, 1). For clarification, we add this detail in the revised caption of Figure 2.
> >
> >
> > ### **7. Formulation of the pre-training weights masking [Q2]**
> > Thank you for pointing out the missing detail. Here, we provide a concrete formulation of the pre-training weights masking operation. Given the pre-training weights $\mathbf{\it{W}} \in \mathbb{R}^{n\*m}$ and masking ratio $r \in (0, 1)$, the masking matrix $\mathbf{\it{M}} \in \{0, 1\}^{n\*m}$ is constructed by:
> > $$
> > \mathbf{\it{M}}\_{i,j} =
> > \begin{cases}
> > 0,      & \mathrm{if} \ |\it{W}\_{i,j}| < \tau\_{r} \\\\
> > 1,      & \mathrm{if} \ |\it{W}\_{i,j}| \geq \tau\_{r}
> > \end{cases}
> > $$
> > , where $\tau_{r}$ is the ${(n\*m\*r)}$-th element in $\\{W_{1},...,W_{n\*m}\\}$ sorted by L1 norm in ascending order. Finally, the masked pre-training weights $\mathbf{\it{\hat{W}}}$ can be formulated as:
> > $$
> > \mathbf{\it{\hat{W}}} = \mathbf{\it{W}} \circ\mathbf{\it{M}}.
> > $$
> > Specifically, we utilize the [l1_unstructured](https://pytorch.org/docs/stable/generated/torch.nn.utils.prune.l1_unstructured.html) function in PyTorch to mask the pre-training weights. For clarification, we add this formulation in Appendix B of the updated manuscript.
> >
> >
> > ### **8. Details of hyper-parameter tuning**
> > In practice, we determine the splitting hyper-parameter $\gamma$ with the linear classifier fine-tuned on low-dimensional representations of downstream data using pre-trained models (line 483-485). In addition, we clarify that *Time* in Figure 1 and Table 1 includes the time of scoring estimation, under-sampling, and the associated hyper-parameter tuning (line 842).
> >
> >
> > [1] Zhou, Daquan, et al. Dataset quantization. ICCV, 2023.
> >
> > [2] Max Welling. Herding dynamical weights to learn. ICML, 2009.
> >
> > [3] Ozan Sener and Silvio Savarese. Active learning for convolutional neural networks: a core-set approach. ICLR, 2018.
> >
> > [4] Sharat Agarwal,et al. Contextual diversity for active learning. ECCV, 2020.

---

> ### Comment · Reviewer_aZeo · 2024-11-20
>
> I want to express thanks to the authors. Most of my concerns have been addressed.
>
> Yet I still cannot understand the incorporation of the learning path very clearly. In the actual implementation, which is claimed to be computed efficiently, the classification losses are averaged. Although the concept of the learning path is interesting, it is not utilized in the implementation. I understand the average can be one of the implementations for the learning path, but it substantially weakens the significance of this concept. It can simply be replaced with models with different capabilities. In addition, you can still use the average of losses from different stages to reflect the learning path.
>
> I suppose the learning path is one of the core ideas in this paper. If the authors can give a better utilization of the learning path in the implementation, I will increase the score.

---

> > ### Author Response · Authors · 2024-11-20
> > **Utilization of the learning path**
> >
> > Thank you for the timely response and we are glad that most concerns have been addressed. There might be some misunderstandings about the learning path. Simply speaking, we use the concept of *learning path* to help readers understand why we need to build models with varying capacities. In our method, we propose to use masked models with varying capacities to establish the learning path. And, we show that previous optimization methods can be also treated as an implementation of the learning path, using models at varying epochs. In Figure 2, we show the ranking correlation between the two implementations: masking and optimization. Compared to the learning path using optimization, our method with masking does not require backpropagations with pretrained models, thereby reducing the computational cost significantly. With the framework, readers can easily understand the motivation of our proposed score - DLC.
> >
> > In addition, we clarify that the average losses of varying models is **an approximation for the definite integral of losses along the learning path**. Thus, it does not weaken the significance of the learning path. Instead, the approximation enables our method to be efficient and practical, which is appreciated by reviewers N2di and FdCJ,
> >
> > We look forward to your response and are willing to answer any questions.

---

> > > ### Comment · Reviewer_aZeo · 2024-11-20
> > >
> > > Thanks for the explanation. I now understand that the formulation does calculate the integral. I will raise the score to 8. But at the same time, I hope the authors can further refine the connection between the motivation and the method design. The motivation is only mentioned once in the abstract. The "easy samples require fewer parameters to learn" leading to the integral of losses along the learning path" will help readers understand the idea better.
> > >
> > > Overall it is an interesting idea with strong experimental support. I'll recommend acceptance.

---

> > > > ### Author Response · Authors · 2024-11-20
> > > > **Many thanks!**
> > > >
> > > > Great thanks for your recognition. We are glad that our explanation addressed your concerns, which also improves the quality of this work. As you suggested, we will improve the writing of motivation to make it clear for readers, in the final version.

---

### Official Review · Reviewer_FdCJ · 2024-11-04

**Soundness:** 2
**Presentation:** 2
**Contribution:** 2
**Rating:** 6
**Confidence:** 3

**Summary:**

A novel training-free hardness score, Distorting-based Learning Complexity, is proposed to identify informative images and instructions from downstream dataset. Also, a flexible under-sampling method with randomness named FlexRand is proposed to alleviate the severe subset distribution shift. Extensive experiments demonstrate the effectiveness and efficiency of the proposed approach.

**Strengths:**

The proposed scoring function, Distorting based Learning Complexity, is an efficient training-free score for dataset pruning. A under-sampling strategy with randomness, FlexRand, is designed to adapt to different data regimes and avoid distribution shift. Extensive experiments demonstrate the effectiveness of the proposed approach. DLC significantly reduces the pruning time by 35× in images pruning benchmark.

**Weaknesses:**

Some typo: Line 021, "a flexible under-sampling with randomness" -> "a flexible under-sampling strategy with randomness"
In Figure 4(a), the MMD value of Random is missing.

**Questions:**

When referring masking the pre-training weights, what specific operation is performed on the network parameters?
What's the meaning of dotted line in blue(10%), green(20%) and orange(30%) in Figure(d)?

---

> ### Author Response · Authors · 2024-11-19
> **Response to Reviewer FdCJ**
>
> Thank you for the positive and constructive feedback. Please find our response below:
>
> ### **1. Typos [W1]**
> Thank you for pointing out the typos. We have fixed these in the revised version.
>
> ### **2. Formulation of the pre-training weights masking [Q1]**
> Thank you for pointing out the missing details. Here, we provide a concrete formulation of the pre-training weights masking operation. Given the pre-training weights $\mathbf{\it{W}} \in \mathbb{R}^{n\*m}$ and masking ratio $r \in [0, 1)$, the masking matrix $\mathbf{\it{M}} \in \{0, 1\}^{n\*m}$ is constructed by:
>
> $$
> \mathbf{\it{M}}\_{i,j} =
> \begin{cases}
> 0,      & \mathrm{if} \ |\it{W}\_{i,j}| < \tau\_{r} \\\\
> 1,      & \mathrm{if} \ |\it{W}\_{i,j}| \geq \tau\_{r}
> \end{cases}
> $$
>
> where $\tau\_{r}$ is the $({n\*m\*r})$-th element in $\{W\_{1},...,W\_{n\*m}\}$ sorted by L1 norm in ascending order. Finally, the masked pre-training weights $\mathbf{\it{\hat{W}}}$ can be formulated as:
> $$
> \mathbf{\it{\hat{W}}} = \mathbf{\it{W}} \circ\mathbf{\it{M}}.
> $$
> In our implementation, we utilize the [l1_unstructured](https://pytorch.org/docs/stable/generated/torch.nn.utils.prune.l1_unstructured.html) function in PyTorch to mask the pre-training weights. For clarification, we add this formulation in Appendix B of the updated manuscript.
>
>
> ### **3. Meaning of dotted lines in Figure 6(d) [Q2]**
> Thank you for pointing out the ambiguous description. In Figure 6(d), each dotted line denotes the results of preserving different percentages of data. In particular, the blue/green/orange dotted lines present the downstream classification accuracy with varying $\gamma$, when preserving 10\%/20\%/30\% data respectively. For clarification, we add the above description in the caption of updated Figure 6(d).

---

### Official Review · Reviewer_N2di · 2024-11-04

**Soundness:** 2
**Presentation:** 2
**Contribution:** 2
**Rating:** 6
**Confidence:** 3

**Summary:**

The paper introduces Distorting-based Learning Complexity (DLC), a novel training-free hardness score for efficient downstream dataset pruning. DLC quantifies sample hardness by masking pre-trained weights and approximating loss integration via the Monte Carlo method. The authors also propose FlexRand, a flexible under-sampling strategy to adapt to different data regimes and avoid distribution shift.

**Strengths:**

1.) The significance lies in its potential to reduce the computational burden of fine-tuning large pre-trained models while maintaining performance.
2.) The paper is well-structured, and easy to follow.
3.) The introduction of FlexRand adds another layer of adaptability to the pruning process, making it more robust across different data regimes, a valuable contribution to data pruning strategies.

**Weaknesses:**

1.) The paper suggests that DLC is not sensitive to the quality of pre-trained models, but this claim could be further experimented on different size level pre-trained models.
2.) The method requires storing multiple masked models, which could be a limitation in environments with constrained memory resources, potentially affecting the practicality of the approach.
3.) The paper could benefit from a more detailed discussion on scenarios where DLC might underperform or fail, providing a more comprehensive understanding of its limitations.

**Questions:**

see the Weaknesses

---

> ### Author Response · Authors · 2024-11-19
> **Response to Reviewer N2di**
>
> Thank you for your positive and valuable suggestions. Please find our response below:
>
>
> ### **1. Results on pre-trained models with various sizes [W1]**
> Thank you for the suggestion. Indeed, we have employed pre-trained models with various sizes (including RN18, RN50, ViT-S, and ViT-B) in the main experiments. In Table 1, we present the results of models pre-trained by fully-supervised learning. In Appendix B.2.3, we present the results of models pre-trained by weakly-supervised learning. These results demonstrate the effectiveness of our method in different-sized pre-trained models.
>
>
> ### **2. Memory for multiple masked models storage [W2]**
> Thank you for pointing out the mistake in the limitation. We clarify that our method does not require storing all the masked models during the pruning. Instead, we dynamically generate the masked model using the [l1_unstructured](https://pytorch.org/docs/stable/generated/torch.nn.utils.prune.l1_unstructured.html) function in PyTorch. In particular, this function enables to pruning of a specific fraction of parameters for a given model. Hence, we calculate the outputs for all data points using the masked models sequentially and do not load the models simultaneously. Despite the sequential operation, our method only requires 1/35 of the computational times of previous methods, as shown in Table 1. To avoid any misunderstanding, we update the limitation in the revised version.
>
>
> ### **3. Discussion of potential failure cases [W3]**
> Thank you for the great suggestion. We conjecture that the effectiveness of our method relies on the model's capability. In particular, our method may fail to improve the performance if the pre-trained model cannot provide high-quality representations. To validate this, we conduct experiments with a ResNet-18 model pre-trained on a small dataset -- CIFAR-10. Obviously, the pre-trained model will perform poorly in producing the representations. We keep the same fine-tuning setting as in the manuscript and report the average classification accuracy over 9 pruning ratios.
>
> As shown in the Table below, our method cannot outperform random selection in this case. Thus, the effectiveness of our method might be limited by the capability of pre-trained models (might be due to the small dataset). We add this discussion in the revised limitations.
>
> |        | CXRB10     | DeepWeeds     | DTD       | FGVCAircraft | Sketch     | Average  |
> | ------ | ------     | ---------     | -----     | ------------ | ------     | ------   |
> | Random | 18.43      | 63.28         | 33.62     | 6.73         | 13.56      | 27.12    |
> | Ours   | 18.61      | 63.38         | 33.75     | 6.62         | 13.65      | 27.20    |

---

> > ### Comment · Reviewer_N2di · 2024-11-28
> >
> > Thank you for your rebuttal, it has addressed the bulk of my concerns.

---

> > > ### Author Response · Authors · 2024-11-28
> > > **Many thanks!**
> > >
> > > Thanks for your recognition. We are glad that our rebuttal addressed your concerns, which also improves the quality of this work.

---

### Author Response · Authors · 2024-11-19
**General Response**

We thank all the reviewers for their time, insightful suggestions, and valuable comments. We are glad that the reviewer (N2di) recognizes the **significance** of downstream dataset pruning. We are also encouraged that reviewers find the method is **effective** (FdCJ,aZeo) on the **extensive** and **detailed** (FdCJ,aZeo) benchmarks, the DLC score is **efficient** (FdCJ), **interesting and practical** (aZeo), and the FlexRand strategy is **robust** and **valuable** (N2di). Besides, reviewers appreciate that the writing is **well-structured**, **fluent**, and **easy to follow** (N2di,aZeo).

In the following responses, we have addressed the reviewers' comments and concerns point by point. The reviews allow us to strengthen our manuscript and the changes$^1$ are summarized below:
- Removed the statement about memory for multiple masked models storage in **Limitations**. [N2di]
- Added discussion of potential failure cases in **Limitations**. [N2di]
- Added formulation of the pre-training weights masking in **Line 199** and **Appendix B**. [FdCJ,aZeo]
- Added comparison of different under-sampling strategies in **Line 457-458** and **Appendix C.2.4**. [aZeo]
- Fixed discussion of pre-trained model quality in **Line 509-517**. [aZeo]
- Added performance of pre-trained models by linear probing in **Line 513** and **Appendix C.2.5**. [aZeo]
- Fixed typos in **Line 021,129,365,487**, **Figure 2**, **Figure 4(a)**, and **Figure 6**.
---
$^1$ For clarity, we highlight the revised part of the manuscript in **blue** color.

---

### Meta-Review · Area_Chair_gtE5 · 2024-12-19

**Metareview:**

The paper proposes a dataset pruning method based on a training-free hardness score called Distorting-based Learning Complexity (DLC) and a flexible under-sampling strategy. Extensive experimental results demonstrate the effectiveness and efficiency of this approach. Overall, the idea is novel, and the performance is impressive. However, the presentation of the motivation can be improved to better articulate the rationale behind the proposed method. Additionally, further clarification would enhance the overall understanding and impact of the work. Based on the overall quality of the work, the novelty of the approach, and the positive feedback from the reviewers, the decision is to recommend the paper for acceptance. We encourage the authors to address the noted shortcomings in the presentation of the motivation in future revisions.

**Additional Comments On Reviewer Discussion:**

The paper was reviewed by three experts in the field and finally received all positive scores: 6, 8, and 6.
The major concerns of the reviewers are:
1.	some details of the method,
2.	additional experimental results to support some claims,
3.	clarification of the motivation,
4.	typo.

The authors address all the above concerns during the discussion period. Hence, I make the decision to accept the paper.

---

### Decision · Program_Chairs · 2025-01-22

Accept (Poster)